# Neutral amino acid transporter SLC38A2 protects renal medulla from hyperosmolarity-induced ferroptosis

**Chunxiu Du[1,2,3,4]\*, Hu Xu[1], Cong Cao[1], Jiahui Cao[1], Yufei Zhang[1], Cong Zhang[1], Rongfang Qiao[1], Wenhua Ming[1], Yaqing Li[1], Huiwen Ren[1], Xiaohui Cui[1], Zhilin Luan[1], Youfei Guan[1,2,3]\*, Xiaoyan Zhang[4]\***

[1]Advanced Institute for Medical Sciences, Dalian Medical University, Dalian, China; [2]Department of Physiology and Pathophysiology, School of Basic Medical Sciences, Dalian Medical University, Dalian, China; [3]Dalian Key Laboratory for Nuclear Receptors in Major Metabolic Diseases, Dalian, China; [4]Health Science Center, East China Normal University, Shanghai, China

**\*For correspondence:**
chunxiu_du@163.com (CD);
youfeiguan@163.com (YG);
xyzhang@hsc.ecnu.edu.cn (XZ)

**Competing interest:** The authors declare that no competing interests exist.

**Abstract** Hyperosmolarity of the renal medulla is essential for urine concentration and water homeostasis. However, how renal medullary collecting duct (MCD) cells survive and function under harsh hyperosmotic stress remains unclear. Using RNA-Seq, we identified SLC38A2 as a novel osmoresponsive neutral amino acid transporter in MCD cells. Hyperosmotic stress-induced cell death in MCD cells occurred mainly via ferroptosis, and it was significantly attenuated by SLC38A2 overexpression but worsened by *Slc38a2*-gene deletion or silencing. Mechanistic studies revealed that the osmoprotective effect of SLC38A2 is dependent on the activation of mTORC1. Moreover, an in vivo study demonstrated that *Slc38a2*-knockout mice exhibited significantly increased medullary ferroptosis following water restriction. Collectively, these findings reveal that *Slc38a2* is an important osmoresponsive gene in the renal medulla and provide novel insights into the critical role of SLC38A2 in protecting MCD cells from hyperosmolarity-induced ferroptosis via the mTORC1 signalling pathway.

## Editor's evaluation

These are nicely done and very complete studies which provide strong evidence that medullary collecting duct cells survive periodic medullary toxicity through upregulation of SLC38A2 , an amino acid transporter which provides substrate for glutathione production in the cells, which is in turn, protective. Cell lines, primary cultures and whole animals are used effectively to strengthen the mechanistic narrative.

## Introduction

The kidney is a central organ that maintains body water and electrolyte homeostasis by producing urine, which depends on the establishment and maintenance of osmotic pressure gradient in the renal medulla. In the human kidney, the osmotic pressure of the inner medulla can reach up to 1200 mOsm, which is fourfold higher than that in the plasma. Renal medullary cells, particularly inner medullary collecting duct (IMCD) epithelial cells, must adapt to constant changes in the osmolarity of their environment. Under dehydrated conditions, IMCD cells are exposed to hyperosmotic stress, which frequently causes cell injury and death. IMCD cells must adapt to and withstand external hyperosmolarity to maintain cell viability and fulfil their functions. It has been previously reported that under

hyperosmotic conditions, renal medullary cells undergo various forms of cell death, such as necrosis and apoptosis (*Han et al., 2011*; *Alfieri et al., 2002*). However, the major form of cell death responsible for hyperosmolarity-induced renal medullary cell injury remains largely unknown. In addition, multiple mechanisms have been demonstrated to contribute to the maintenance of cell viability of renal medullary cells under dehydrated conditions. Among these, the transcription factor tonicity-responsive enhancer binding protein (TonEBP) and its target osmoprotective genes, including aldose reductase and heat shock protein 70 (HSP70), represents the most important protective mechanisms (*Lee et al., 2011*). Other factors, including nuclear factor kappa B (NF-κB), farnesoid X receptor (FXR), peroxisome proliferator-activated receptor β/δ (PPARβ/δ), and cyclooxygenase 2 (COX-2), have also been reported to contribute to the survival of renal medullary cells in a hyperosmotic environment (*Hasler et al., 2008*; *Xu et al., 2018*; *Guan, 2004*; *Moeckel et al., 2003*). However, the underlying mechanism by which hyperosmolarity induces medullary cell injury and cell survival under hyperosmotic stress remains incompletely understood and requires further investigation.

Regulated cell death (RCD) is required for the normal development and maintenance of organ homeostasis. RCD signalling cascades, including apoptosis, necroptosis, pyroptosis, ferroptosis, and autophagy-associated cell death, are tightly regulated to establish a functional tissue architecture (*Santagostino et al., 2021*). Unlike most other cells that live under isosmotic conditions, medullary collecting duct (MCD) cells reside in an environment of variable hyperosmolarity, which frequently causes cell damage and even death. Ferroptosis is a novel type of RCD that differs from apoptosis, pyroptosis, cell necrosis, and autophagy (*Li et al., 2020*). It is driven by iron-dependent oxidative destruction of the lipid bilayer and is caused by an imbalance of oxidation and antioxidation in cells (*Yan et al., 2021*). Emerging evidence suggests that hyperosmolarity results in a robust increase in reactive oxygen species (ROS) production (*Yang et al., 2005*; *Ikari et al., 2013*; *Ma et al., 2019*), suggesting that ferroptosis may contribute to cell damage and death of renal medullary cells under dehydration conditions. However, the role and mechanism of ferroptosis in the survival of MCD and interstitial cells under hyperosmotic conditions are currently unclear.

Unrestrained lipid peroxidation is a hallmark of ferroptosis. The reduction of lipid peroxidation to hydroxyl by glutathione peroxidase4 (GPX4) requires the catalytic selenocysteine residue of GPX4 and two electrons provided mainly by glutathione (GSH) (*Jiang et al., 2021*). Cysteine (Cys) is the rate-limiting substrate in the biosynthesis of reduced GSH. Deprivation of cystine or Cys or inhibition of their membrane transporters (system Xc⁻ cystine/glutamate antiporter) induces ferroptosis by depleting cellular GSH and increasing ROS (*Yang et al., 2022*). Glutamine (Gln) metabolism is tightly linked to ferroptosis regulation. After influx into cells via SLC1A5 (ASCT2), Gln is converted to glutamate (Glu) to produce GSH, together with Cys and glycine (Gly), to maintain redox homeostasis (*Hensley et al., 2013*). Collectively, these findings demonstrate that dysregulation of amino acid metabolism plays an important role in the regulation of ferroptosis (*Jiang et al., 2021*; *Cormerais et al., 2020*). It is anticipated that sodium-dependent neutral amino acid transporter-2 (*Slc38a2*), as a Cys, Glu, Gly, and methionine (Met) transporter and an osmoresponsive gene, may have a great impact on GPX4 activity and GSH biosynthesis (*Nahm et al., 2002*; *Mackenzie and Erickson, 2004*). However, whether SLC38A2 contributes to ferroptosis regulation in renal medullary cells under hyperosmotic stress has not been explored and warrants further investigation.

In the present study, we found that ferroptosis is the major RCD in IMCD cells exposed to hyperosmotic stress. Using RNA-Seq, we found that hyperosmolarity induced *Slc38a2* expression in IMCD cells in an NF-κB-dependent manner. Using *Slc38a2*-knockout mice and cultured IMCD cells, we further demonstrated that SLC38A2 exerted an osmoprotective effect on IMCD cells by inhibiting ferroptosis through activation of the SLC38A2-mTORC1 pathway. These findings uncover SLC38A2 as an osmoprotective amino acid transporter that protects renal MCD cells by inhibiting hyperosmolarity-induced ferroptosis via mTORC1 activation. *Slc38a2* is a critical osmoprotective gene in the renal medulla, which is a prerequisite for urine concentration and water homeostasis.

## Results

### Ferroptosis is the major type of regulated cell death in IMCD cells exposed to hyperosmolarity

Multiple types of RCD are involved in hyperosmolarity-induced cell damages, including apoptosis, necroptosis, and autophagy (*Xu et al., 2018*; *Nunes et al., 2013*; *Bittner et al., 2017*). We and others have previously reported that hyperosmotic stress significantly decreases renal medullary cell viability and induces apoptosis in only a small portion of the cell population (*Xu et al., 2018*; *Dmitrieva et al., 2000*; *Küper et al., 2011*), suggesting that other types of RCD may account for hyperosmolarity-elicited cell death. To characterise the exact RCD type responsible for hyperosmolarity-induced cell death and identify new mechanistic pathways involved in the osmoprotection of renal medullary cells, we treated mIMCD3 cell line with hyperosmotic medium for 12 h. As previously reported, hyperosmolarity significantly reduced cell viability. Ultrastructural analysis by transmission electron microscopy revealed that hyperosmolarity-treated mIMCD3 cells exhibited typical morphological features of ferroptosis, including shrunken mitochondria with disappeared and fragmented cristae and disrupted cell membranes, suggesting that ferroptosis is involved in hyperosmolarity-induced cell death (*Figure 1—figure supplement 1*). To prove this, we performed whole transcriptome analysis using RNA-Seq in mIMCD3 cells incubated under either isosmotic or hyperosmotic conditions and found that the expression of many genes associated with ferroptosis, including *Slc7a11*, *Slc40a1*, *Gpx4*, *Acsl4*, *Vdac2*, *Tfrc*, *Ncoa4*, and *Slc3a2*, was significantly altered after hyperosmotic treatment (*Figure 1—figure supplement 2*). To confirm the involvement of ferroptosis in hyperosmolarity-induced cell death, we pre-treated mIMCD3 cells with liproxstatin-1 (Lip-1), an inhibitor of ferroptosis, followed by hyperosmotic stress. The result showed that Lip-1 attenuated cell viability in a dose-dependent manner (*Figure 1A*). Cell viability was also improved in a dose-dependent manner by ferrostatin-1 (Fer-1), another ferroptosis inhibitor (*Figure 1B*). Lip-1 and Fer-1 at high doses almost completely abolished hyperosmolarity-induced cell death, suggesting that ferroptosis is the major type of RCD contributing to hyperosmotic stress-induced cell injury in IMCD cells (*Figure 1A and B*). In support of this, a cytometry study showed that hyperosmolarity significantly increased cellular lipid ROS levels, which were completely suppressed by the ferroptosis inhibitors Lip-1 and Fer-1, indicating that hyperosmolarity promotes ferroptosis mainly by enhancing lipid peroxidation (*Figure 1C*). Conversely, the ferroptosis inducers, erastin and RSL-3, markedly increased the osmosensitivity of mIMCD3 cells in a dose-dependent manner (*Figure 1D and E*). Consistently, both erastin and RSL3 aggravated hyperosmolarity-induced lipid ROS accumulation in IMCD cells (*Figure 1F*). We also noticed that RSL3 caused a comparable degree of cell death and lipid ROS production between isosmolarity- and hyperosmolarity-treated mIMCD3 cells, suggesting that GPX4 activity is critical for the survival of mIMCD3 cells. Altogether, these findings demonstrate that ferroptosis is the major type of RCD in IMCD cells exposed to hyperosmolar stress.

### Slc38a2 is an osmoresponsive gene in cultured mIMCD3 cells

RNA-Seq analysis showed that 5197 genes were upregulated, and 4069 genes were downregulated after hyperosmotic treatment (*Figure 2A and B*). Among these were numerous osmoresponsive genes, including *Hspa1a*, *Hspa1b*, *Hspa41*, *Nfat5* (*TonEBP*), and *Hif1α*, which have been previously reported to be induced by hyperosmolarity (*Yang et al., 2005*; *Neuhofer et al., 2004*; *Figure 2C*). As the solute carrier (SLC) family plays an important role in the survival of renal IMCD cells under hyperosmolar conditions, we systemically analysed the mRNA levels of the *Slc* family and found that the expression of *Slc38a2*, *Slco4a1*, *Slc5a3*, *Slc34a2*, *Slc45a3*, *Slc30a4*, *Slc15a3*, *Slc7a11*, *Slc6a6*, and *Slc16a10* was significantly upregulated under hyperosmotic conditions (*Figure 2—figure supplement 1A*). As emerging evidence suggests that amino acids are important osmolytes in the regulation of osmoadaptive responses (*Bevilacqua et al., 2005*; *Horio et al., 1997*; *Dall'Asta et al., 1999*), we further examined the presence of each *Slc* family member response for amino acid transport in the kidney using publicly available single-cell RNA-Seq databases (https://www.bioparadigms.org/, https://cello.shinyapps.io/kidneycellexplorer/). The results showed that 51 amino acid transporters were expressed in mouse kidneys, where *Slc38a2* represents one of the major amino acid transporters with high medullary tubule expression (*Figure 2—figure supplement 2*). We further found that among the 51 amino acid transporters expressed in the kidney, *Slc38a2* is a member of the *Slc38* subfamily, with the highest expression level in cultured mIMCD3 cells (*Figure 2—figure supplement*

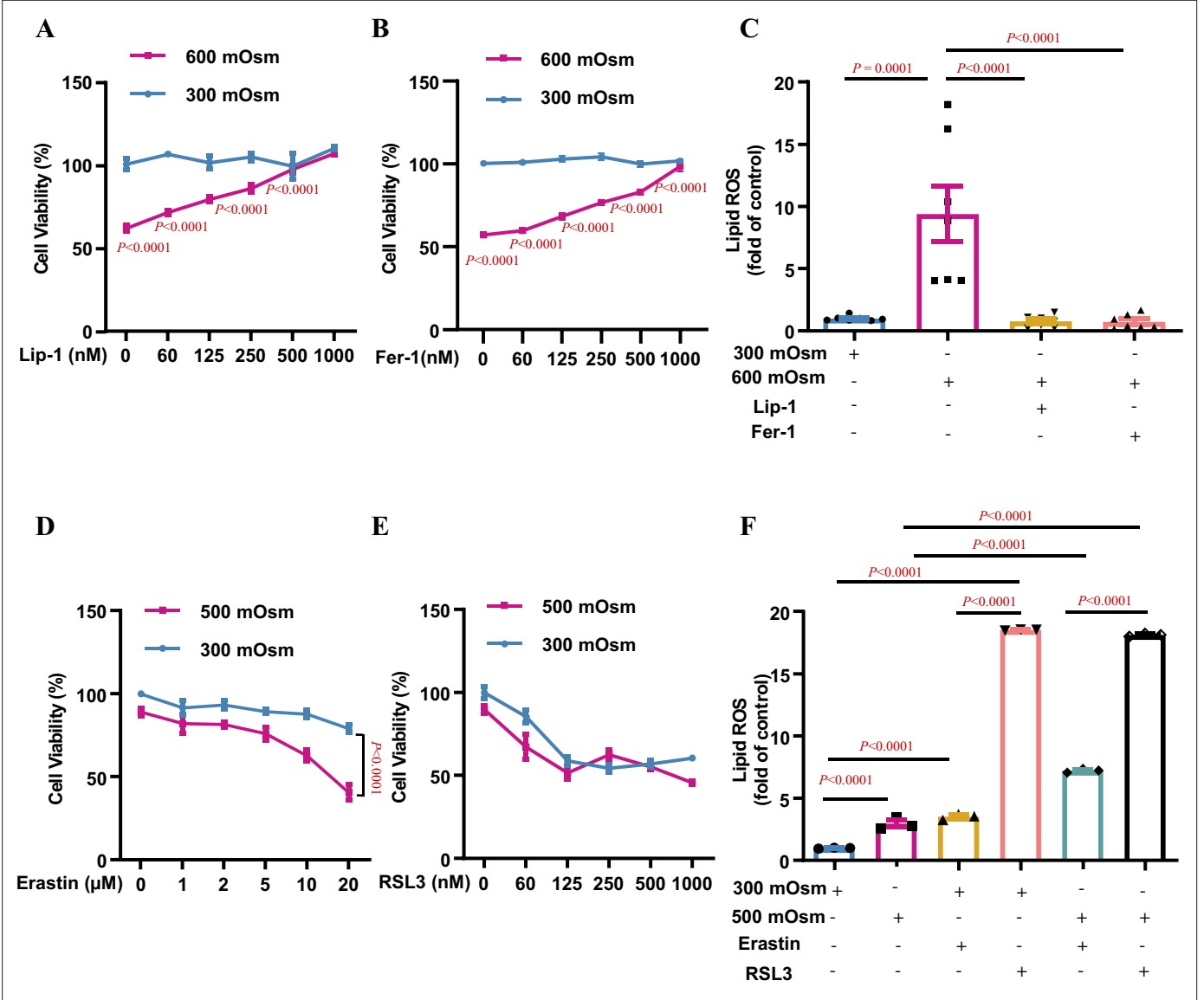

**Figure 1.** Ferroptosis is the major type of regulated cell death in inner medullary collecting duct (IMCD) cells under hyperosmotic stress. mIMCD3 cells were treated with various doses of the ferroptosis inhibitors and inducers for 24 h, with or without hyperosmotic exposure in the final 12 h. (**A and B**) The 3-(4,5-dimethylthiazol-2-yl)-2,5-diphenyltetrazolium bromide (MTT) reduction assay showing the ferroptosis inhibitor liproxstatin-1 (Lip-1) (**A**) and ferrostatin-1 (Fer-1) (**B**) increased the cell viability under hyperosmotic condition (600 mOsm) in a dose-dependent manner. n=4. (**C**) Flow cytometry assay showing that Lip-1 (1 μM) and Fer-1 (1 μM) completely abolished hyperosmolarity-elicited lipid reactive oxygen species (ROS) production. n=7. (**D**) The system Xc⁻ inhibitor erastin worsened hyperosmolarity-induced mIMCD3 cell death in a dose-dependent manner. Note: Erastin slightly reduced the viability of mIMCD3 cells exposed to isosmolarity. n=4. (**E**) The GPX4 inhibitor RSL3 induced mIMCD3 cell death under either isosmotic or hyperosmotic conditions. A slight difference in cell viability was found between the two groups. n=4. (**F**) Flow cytometry analysis results showing that RSL3 caused comparable production of lipid ROS in mIMCD3 cells treated with isosmolarity or hyperosmolarity. Compared to erastin, RSL3 was more potent in promoting lipid ROS production. n=3. Data are means ± SEM; two-tailed Student's *t*-test for A and B; one-way ANOVA tests for C and F; two-way ANOVA tests for D and E. See numerical source data in *Figure 1—source data 1*.

The online version of this article includes the following source data and figure supplement(s) for figure 1:

**Source data 1.** Numerical source data for *Figure 1*.

**Figure supplement 1.** Hyperosmolarity treatment leads to reduced cell viability and severe mitochondrial damage.

**Figure supplement 1—source data 1.** Numerical source data for *Figure 1—figure supplement 1*.

**Figure supplement 2.** Ferroptosis is involved in the hyperosmolality-induced cell death of inner medullary collecting duct cells.

**Figure supplement 2—source data 1.** Numerical source data for *Figure 1—figure supplement 2*.

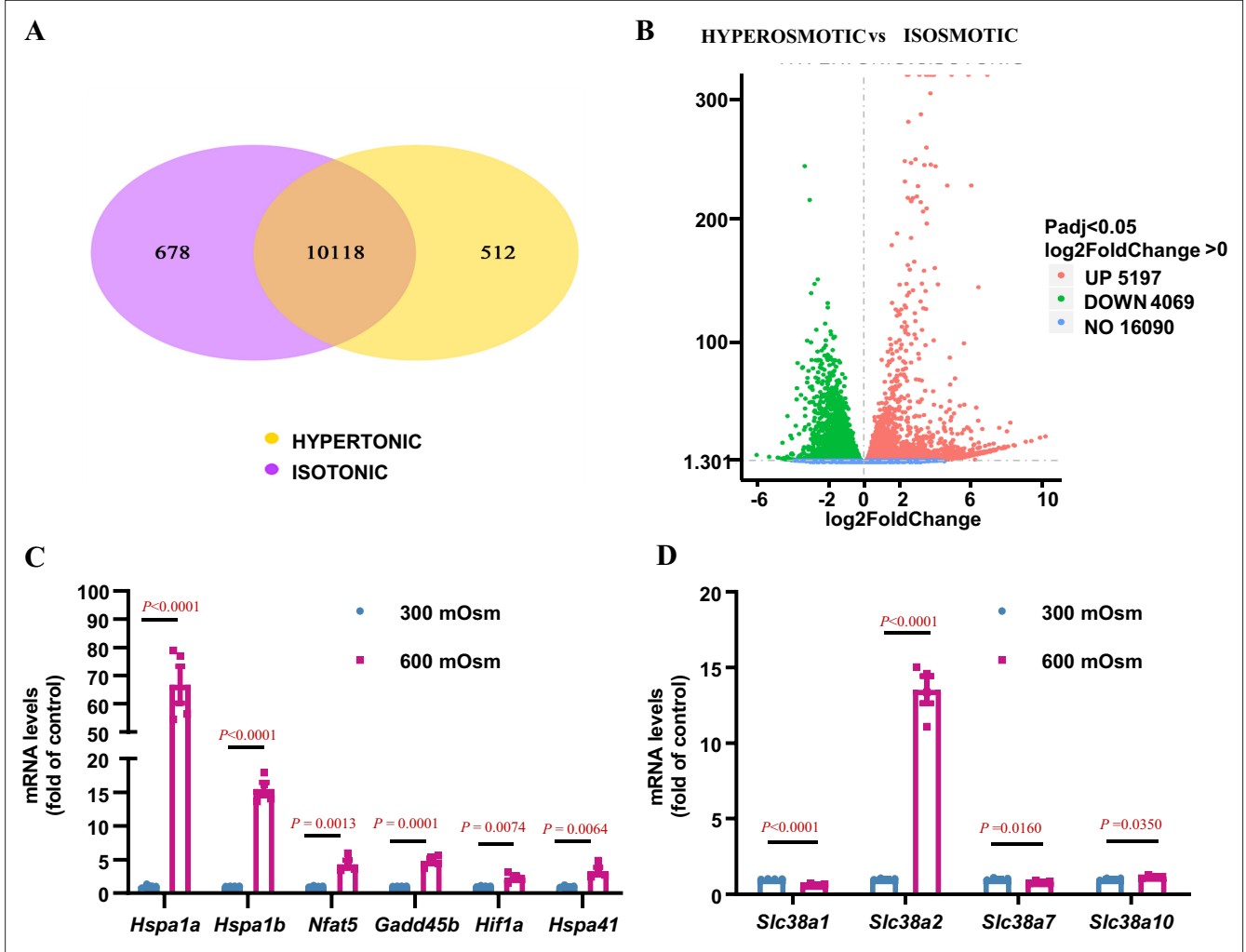

**Figure 2.** RNA-Seq analysis identifies *Slc38a2* as an osmoregulated neutral amino acid transporter in mIMCD3 cells. mIMCD3 cells were cultured under either isosmotic (300 mOsm) or hyperosmotic (600 mOsm) conditions for 6 h, and mRNA was then extracted for RNA-Seq analysis. (**A**) Venn diagram showing the overlap of different genes among the hyperosmotic and isosmotic groups. The co-expressed genes between the hyperosmotic and isosmotic groups are also shown. The sum of all the numbers in the circle of Venn diagram represents the total number of differential genes in the comparison combination, and the overlap area represents the number of common differential genes in the combination. (**B**) The volcano map visually displaying the differential gene distribution of the hyperosmotic and isosmotic groups. The abscissa in the figure represents the fold change of gene expression (log2FoldChange) in the groups, and the ordinate represents the significance level of the difference in gene expression between the treatment and control groups (−log10padj or −log10pvalue). Red dots represent the upregulated genes, and green dots indicate downregulated genes. The volcano map shows that 5197 genes were upregulated, and 4069 genes were downregulated after hyperosmotic treatment. (**C**) RNA-Seq analysis results showing that hyperosmolarity increased the mRNA levels of previously reported osmoregulatory genes. After hyperosmotic treatment, the expression of *Hspa1a*, *Hspa1b*, *Hspa41*, *Nfat5*, *Gadd45b*, and *Hif1a* was significantly increased. n=4. (**D**) RNA-Seq results demonstrating that hyperosmolarity significantly induced *Slc38a2* mRNA expression in mIMCD3 cells. A slight change in *Slc38a1*, *Slc38a7*, and *Slc38a10* expression was observed. n=4. Data are means ± SEM; two-tailed Student's *t*-test for **C–D**. See numerical source data in *Figure 2—source data 1*.

The online version of this article includes the following source data and figure supplement(s) for figure 2:

**Source data 1.** Numerical source data for *Figure 2*.

**Figure supplement 1.** RNA-Seq analysis shows that hyperosmolarity induces *Slc38a2* mRNA expression in mIMCD3 cells.

**Figure supplement 1—source data 1.** Numerical source data for *Figure 2—figure supplement 1*.

**Figure supplement 2.** Expression and distribution of amino acid transporters in the juxtamedullary and cortical nephron and renal collecting duct.

**Figure supplement 3.** RNA-Seq analysis identifies *Slc38a2* as an osmoregulated amino acid transporter in mIMCD3 cells.

**Figure supplement 4.** SLC38A2 protein is expressed in the cell membrane and cytoplasm.

**Figure supplement 4—source data 1.** Numerical and uncropped western blot source data for *Figure 2—figure supplement 4*.

*1B*) and is the major member of the *Slc38* subfamily, whose expression was significantly induced by hyperosmolarity (*Figure 2D*, *Figure 2—figure supplement 1C*, *Figure 2—figure supplement 3*). As expected, *Slc38a2* was mainly expressed in the cell membrane and cytoplasm of 293T cells transfected with an expression vector, carrying an EGFP-fused full-length mouse *Slc38a2* cDNA sequence (*Figure 2—figure supplement 4*). Collectively, these findings demonstrate that *Slc38a2* is a novel osmoresponsive gene in renal MCD cells.

## Intrarenal localisation and osmoregulation of SLC38A2 in the mouse kidney

To study the physiological function of SLC38A2 in the kidney, we first determined the intrarenal localisation of SLC38A2 in the mouse kidney. We searched a single-cell RNA-Seq database (https://cello. shinyapps.io/kidneycellexplorer/) of mouse kidneys and mapped their expression in the nephron. The results showed that *Slc38a2* mRNA was abundantly expressed in the medullary loop of Henle and MCD epithelial cells (*Figure 3—figure supplement 1*). To determine the intrarenal localisation of SLC38A2, an immunohistochemical assay was performed using an SLC38A2-specific antibody. We found that SLC38A2 protein was ubiquitously expressed in most renal tubule segments, with relatively higher levels in renal MCD epithelial cells (*Figure 3A*). These findings demonstrate that SLC38A2 is constitutively expressed in MCD epithelial cells. To determine whether SLC38A2 expression was upregulated under hyperosmotic conditions in vivo, C57BL/6 wild-type (WT) mice were deprived of water for 24 h, and the expression of *Slc38a2* mRNA (*Figure 3B*) and protein (*Figure 3C–D*) was examined. The results showed that both *Slc38a2* mRNA and protein expression were significantly upregulated in renal medullas (*Figure 3B–D*), especially in renal MCD cells, as assessed by immunohistochemical analysis (*Figure 3E*). These results are consistent with the in vitro findings and demonstrate that SLC38A2 is constitutively expressed in renal MCD cells, where its expression is significantly upregulated in the renal medulla of dehydrated mice.

## *Slc38a2* gene deficiency accelerates renal medullary cell ferroptosis in dehydrated mice

To study the role of SLC38A2 protein in the mouse kidney, we used CRISPR/Cas9 to generate a mouse deficient for *Slc38a2* on a C57BL/6 background (*Figure 4A*, *Figure 4—figure supplement 1A and B*). Owing to the high incidence of postnatal lethality in *Slc38a2*-knockout mice (*Slc38a2$^{-/-}$*) on a C57BL/6 background (*Weidenfeld et al., 2021*), we crossed *Slc38a2* heterozygous mice with WT mice on a 129 background for five generations to obtain developmentally normal *Slc38a2*-knockout mice for in vivo studies. Using RT-PCR and western blotting assays, we validated the successful generation of *Slc38a2$^{-/-}$* mice (*Figure 4B–C*). To explore the role of SLC38A2 in the kidney, we collected 24 h urine samples from WT and *Slc38a2$^{-/-}$* mice and found that *Slc38a2$^{-/-}$* mice exhibited a polyuria phenotype (*Figure 4—figure supplement 2A and B*) and increased urine osmolyte excretion (*Figure 4—figure supplement 2C*). After 24 h water restriction, although no difference in urine output and osmolality was observed between the two genotypes, *Slc38a2$^{-/-}$* mice exhibited an increased urine volume to body weight ratio (*Figure 4—figure supplement 2D–F*). These results suggested that SLC38A2 may play an important role in the establishment of a medullary osmotic gradient and urine concentration. To determine whether *Slc38a2* gene deficiency is involved in ferroptosis in the renal medulla of water-deprived mice, we measured $Fe^{2+}$, glutathione, and malondialdehyde (MDA) levels and superoxide dismutase (SOD) activity in the renal medulla of both genotypes. The results showed that *Slc38a2$^{-/-}$* mice exhibited a significant increase in $Fe^{2+}$ and MDA levels and a marked decrease in GSH and SOD concentrations (*Figure 4D–I*). Furthermore, immunohistochemical analysis revealed that *Slc38a2$^{-/-}$* mice exhibited much higher levels of 4-Hydroxynonenal (4-HNE, a well-studied aldehyde product of phospholipid peroxidation) and 8-oxodG (a product of DNA oxidative damage) in the renal medulla than WT mice (*Figure 4J&K*). These findings suggest that *Slc38a2* deficiency is associated with increased iron-dependent lipid peroxidation, a hallmark of ferroptosis in the renal medulla of dehydrated mice. Consistent with the in vitro findings, under water deprivation, *Slc38a2* mice displayed more severe suppression of GPX4 levels and mTORC1 activity than WT mice (*Figure 4L*, *Figure 4—figure supplement 3A*), suggesting that the renal medullary cells of *Slc38a2$^{-/-}$* mice were more susceptible to hyperosmotic stress-induced cell injury. This speculation was further confirmed by the finding that water restriction resulted in a marked increase in cell death in the medulla of

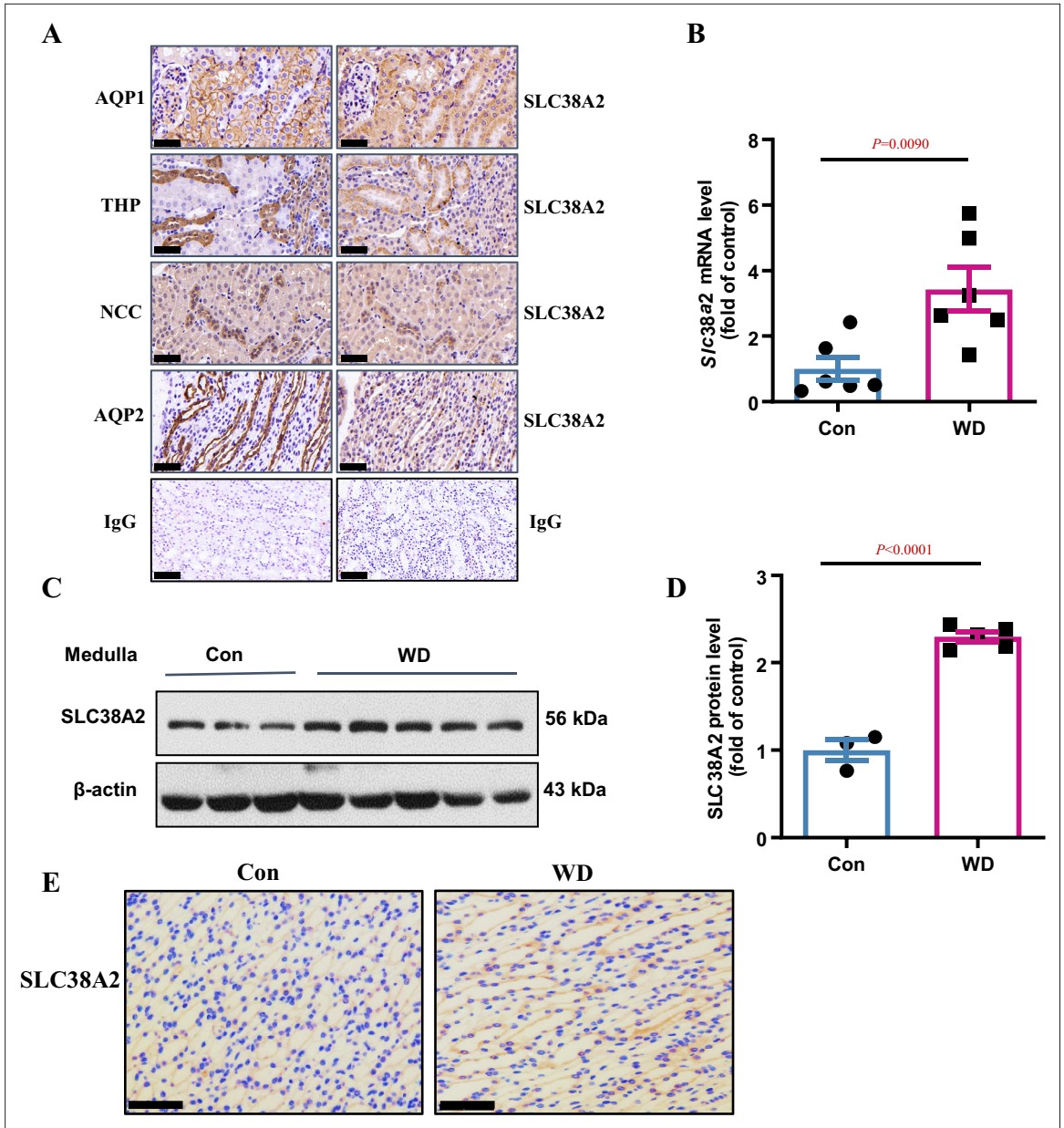

**Figure 3.** Intrarenal localisation and osmoregulation of SLC38A2 protein in the kidney. (**A**) Immunohistochemical analysis results indicating intrarenal localisation of SLC38A2 protein in mice. The results showed that SLC38A2 was expressed at relatively high level in the proximal tubule, thick ascending limb, and collecting duct, with low expression in the distal tubule. AQP1, THP, NCC, and AQP2 are markers for the proximal tubule, distal tubule, thick ascending limb, and collecting duct, respectively. (**B–D**) Water deprivation increased medullary *Slc38a2* mRNA (**B**) and protein (**C**) expression in mice. Male C57BL/6 mice were deprived of water for 24 h. Real-time PCR and western blotting analyses were used to measure *Slc38a2* mRNA and protein expression in renal medullas, respectively. SLC38A2 protein levels were semi-quantitated (**D**). B, n=6; D, n=3–5. (**E**) Immunohistochemical analysis results showing that SLC38A2 protein expression was induced in renal medullary collecting duct cells. Scale bar = 50 µm. WD, water deprivation. Data are means ± SEM. Two-tailed Student's *t*-test for B and D. See numerical source data and uncropped western blot images in *Figure 3—source data 1*.

The online version of this article includes the following source data and figure supplement(s) for figure 3:

**Source data 1.** Numerical and uncropped western blot source data for *Figure 3*.

**Figure supplement 1.** Intrarenal localisation and osmoregulation of *Slc38a2* mRNA in the kidney.

*Slc38a2* $^{-/-}$ mice as assessed by the TUNEL assay, with little change in caspase-3 expression and activity (*Figure 4M–N*, *Figure 4—figure supplement 3B–C*). Collectively, these findings demonstrate that global *Slc38a2* gene deficiency accelerates renal medullary cell ferroptosis in dehydrated mice with damaged urine concentration, suggesting that SLC38A2 expression in the renal collecting duct is

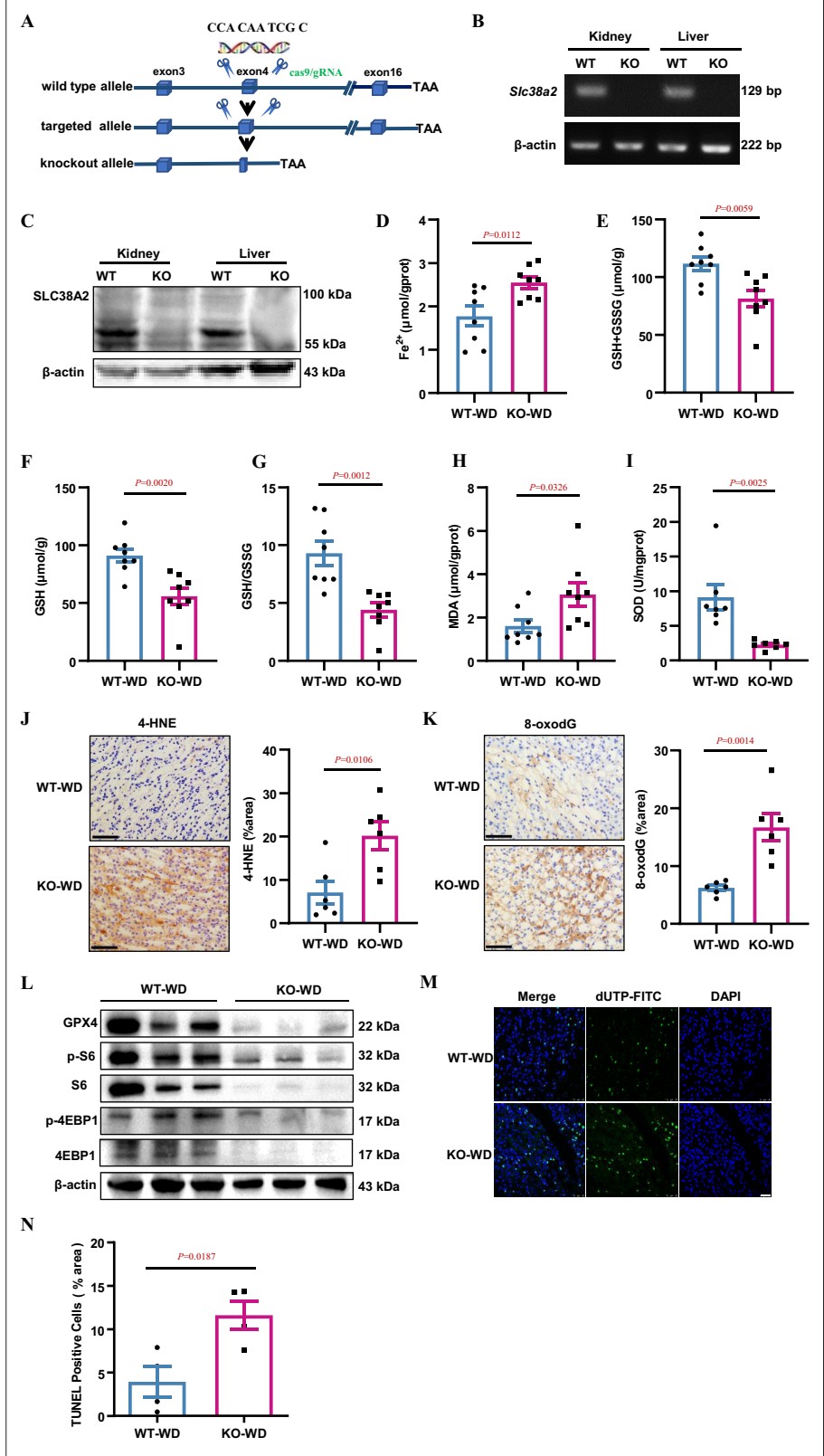

**Figure 4.** *Slc38a2*-gene deficiency accelerates renal medullary cell ferroptosis in dehydrated mice. (**A**) Strategy for generating global *Slc38a2*-gene knockout mice using the CRISP/cas9 technique to delete a 10 bp of DNA sequence (5'-CCA CAA TCG C-3') in the fourth exon of the *Slc38a2* gene. (**B and C**) RT-PCR (**B**) and western blotting (**C**) assays showing the absence of Slc38a2 mRNA and protein in the kidney and liver of *Slc38a2*⁻/⁻ mice,

*Figure 4 continued on next page*

*Figure 4 continued*

respectively. (**D**) *Slc38a2*$^{-/-}$ mice exhibited significantly increased renal medullary Fe$^{2+}$ concentration compared to wild-type mice after water deprivation. n=8. (**E–G**) Water deprivation resulted in a significant decrease in total (**E**) and reduced (**F**) GSH levels and GSH/GSSG ratios in the renal medulla of *Slc38a2*-KO mice. n=8. (**H and I**) Water deprivation resulted in a significant increase in MDA (**H**) and a marked reduction in SOD activity (**I**) in the medulla of *Slc38a2*$^{-/-}$ mice. n=7–8. (**J and K**) Immunohistochemical analysis results showing that 4-HNE and 8-oxo-dG levels were significantly increased in the renal medulla of *Slc38a2*$^{-/-}$ mice compared with that in wild-type mice after water deprivation. n=6. Scale bar = 50 µm. (**L**) Western blotting assay results demonstrating that water deprivation led to a significant decrease in the levels of total and phosphorylated S6 and 4EBP1 protein expression in the medulla of *Slc38a2*$^{-/-}$ mice compared with that in wild-type mice. (**M**) Immunofluorescence analysis results showing that water restriction resulted in a marked increase in cell death (green) in the medulla of *Slc38a2*$^{-/-}$ mice as assessed by the TUNEL assay. Scale bar = 25 µm. (**N**) Water restriction resulted in a marked increase in cell death in the medulla of the *Slc38a2*$^{-/-}$ mice as assessed by the TUNEL assay. n=4. WT, wild-type; KO, *Slc38a2*$^{-/-}$; WD, water deprivation; GSH, reductive glutathione; GSSG, oxidised glutathione; MDA, malondialdehyde; SOD, superoxide dismutase. Data are means ± SEM; two-tailed Student's *t*-test for **D, E, F, G, H, I, J, K, and N**. See numerical source data and uncropped western blot images in *Figure 4—source data 1*.

The online version of this article includes the following source data and figure supplement(s) for figure 4:

**Source data 1.** Numerical and uncropped western blot source data for *Figure 4*.

**Figure supplement 1.** Generation and validation of global *Slc38a2*-gene knockout mice.

**Figure supplement 1—source data 1.** Numerical source data for *Figure 4—figure supplement 1*.

**Figure supplement 2.** Effect of *Slc38a2*-gene deficiency on urine output and osmolality in mice.

**Figure supplement 2—source data 1.** Numerical source data for *Figure 4—figure supplement 2*.

**Figure supplement 3.** Effect of water deprivation on mTORC1 activity and ferroptosis in the renal medulla of *Slc38a2*$^{-/-}$ mice.

**Figure supplement 3—source data 1.** Numerical and uncropped western blot source data for *Figure 4—figure supplement 3*.

critical in maintaining urine concentration capacity, mainly by protecting IMCD cells from ferroptosis under dehydration.

## Hyperosmolarity induces SLC38A2 expression in cultured mIMCD3 cells and primary IMCD cells

To further confirm the induction of SLC38A2 by hyperosmotic stress, we determined the effect of hyperosmolarity on SLC38A2 expression at both the mRNA and protein levels in a cultured mouse IMCD cell line (mIMCD3 cells) (*Figure 5*) and mouse primary IMCD cells (*Figure 5—figure supplement 1*). In mIMCD3 cells, SLC38A2 expression was significantly induced by hyperosmolarity at both the mRNA and protein levels in a dose- (*Figure 5A–C*) and time-dependent manner (*Figure 5D–G*). We found that *Slc38a2* mRNA and protein expression reached their highest levels at 6 and 12 h after hyperosmotic exposure, respectively (*Figure 5D–G*). Immunofluorescence assay further showed that hyperosmolarity increased total SLC38A2 protein expression and its membrane translocation (*Figure 5H*). Similarly, in primary IMCD cells (*Figure 5—figure supplement 1A and B*), hyperosmolarity also induced *Slc38a2* mRNA and protein expression, as assessed using real-time PCR, western blotting, and immunofluorescence assays (*Figure 5—figure supplement 1C–F*). These findings demonstrate that hyperosmolarity induces SLC38A2 expression in renal collecting duct cells, which may contribute to the osmoadaptive response to medullary hyperosmotic stress.

## Hyperosmolarity induces SLC38A2 expression by activating the NF-κB pathway

To elucidate the molecular mechanism by which hyperosmolarity induces SLC38A2 expression, we determined the role of NF-κB in SLC38A2 expression in mIMCD3 cells under hyperosmotic stress. Hyperosmolarity treatment for 12 h increased p-NF-κB expression in a dose-dependent manner, but decreased p-IκBα expression (*Figure 6A*), suggesting an induction of NF-κB activity by hyperosmolality. Immunofluorescence assay revealed that hyperosmolarity promoted the nuclear translocation of NF-κB in mIMCD3 cells (*Figure 6B*). Consistent with the in vitro findings, water deprivation for

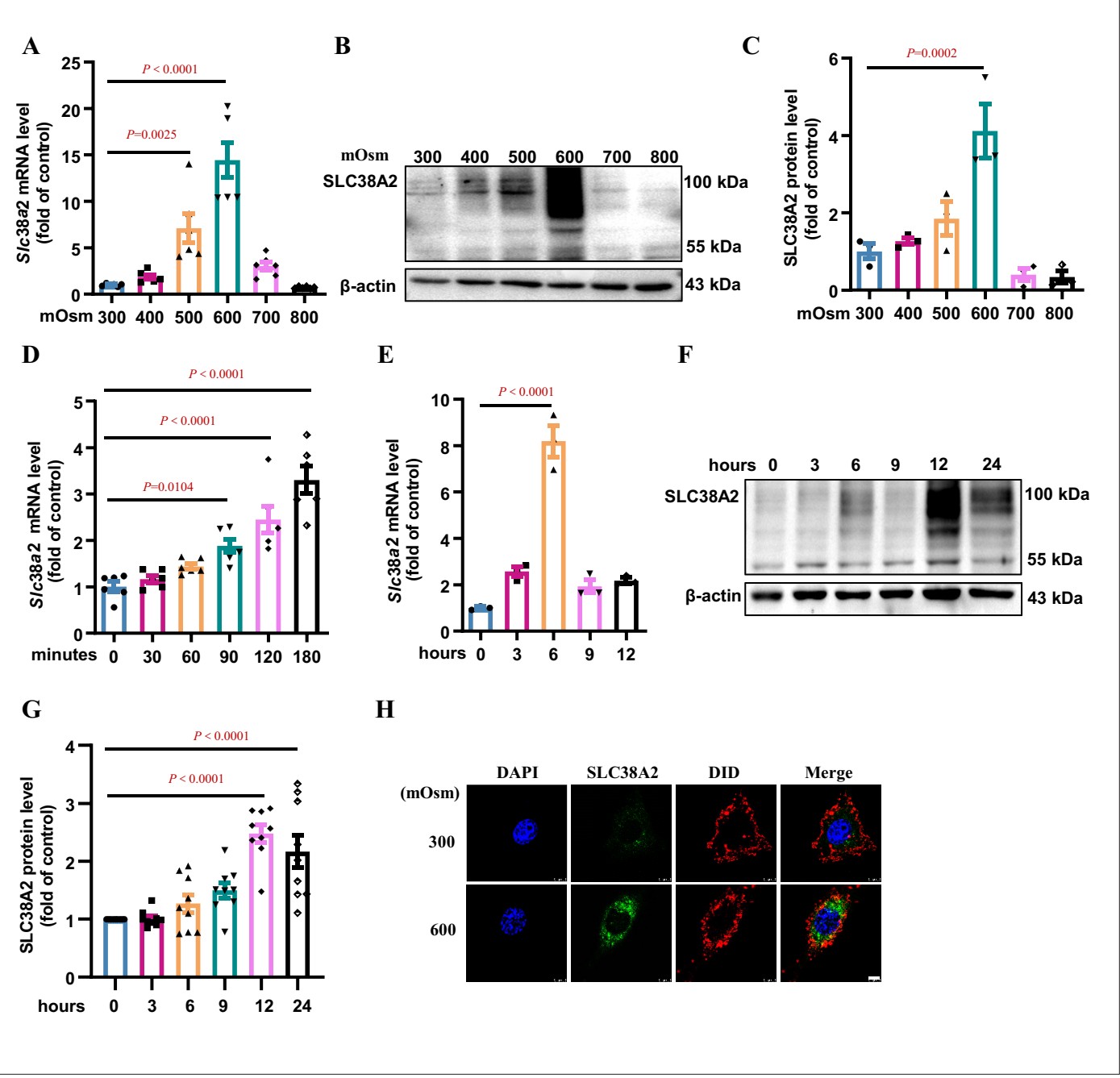

**Figure 5.** Hyperosmolarity induces SLC38A2 expression in mIMCD3 cells. (**A**) Real-time PCR assay showing that hyperosmolarity induced *Slc38a2* mRNA expression in mIMCD3 cells in a dose-dependent manner. The cells were incubated with hyperosmotic solutions (400–800 mOsm) for 6 h. n=6. (**B–C**) Western blotting analysis demonstrating that hyperosmolarity induced SLC38A2 protein expression in mIMCD3 cells in a dose-dependent manner. The cells were exposed to various degrees of hyperosmotic stress (400, 500, 600, 700, and 800 mOsm) for 12 h. SLC38A2 protein expression levels were quantitated (**C**). n=3. (**D–E**) Time-dependent effect of hyperosmolarity on *Slc38a2* mRNA expression. The cells were treated with a hyperosmotic solution (600 mOsm) at various time intervals. *Slc38a2* mRNA expression began to increase at 1.5 h (**D**) and reached the highest levels at 6 h after hyperosmolarity treatment (**E**). n=3–6. (**F–G**) Time-dependent induction of SLC38A2 protein expression by hyperosmolarity. SLC38A2 protein expression began to increase at 6 h (**F**) and reached the highest level at 12 h after the treatment (**G**). n=9. (**H**) Immunofluorescence studies showing that hyperosmolarity markedly induced SLC38A2 expression. DAPI stains the nucleus in blue. DID stains the cell membrane in red. SLC38A2 protein was stained in green. Scale bar = 5 µm. Data are means ± SEM; one-way ANOVA tests for A, C, D, E, and G. DAPI, 4',6-diamidino-2-phenylindole; DID, 1,1'-dioctadecyl-3,3,3',3'-tetramethylindodicarbocyanine,4-chlorobenzenesulfonate salt. See numerical source data and uncropped western blot images in *Figure 5—source data 1*.

The online version of this article includes the following source data and figure supplement(s) for figure 5:

*Figure 5 continued on next page*

*Figure 5 continued*

**Source data 1.** Numerical and uncropped western blot source data for *Figure 5*.

**Figure supplement 1.** Hyperosmolarity induces SLC38A2 expression in mouse primary inner medullary collecting duct (IMCD) cells.

**Figure supplement 1—source data 1.** Numerical and uncropped western blot source data for *Figure 5—figure supplement 1*.

24 h, which markedly increased renal medullary osmolality, significantly upregulated p-NF-κB protein levels and the ratios of p-NF-κB to total NF-κB in the renal medulla of WT C57BL/6 mice (*Figure 6C*). We further analysed the mouse *Slc38a2* gene promoter and found a potential NF-κB-binding site in the sequence between –867 nt and –876 nt (*Figure 6D*). Luciferase reporter assay showed that NF-κB promoted mouse *Slc38a2* gene promoter activity (*Figure 6D*), and point mutation experiments further defined the sequence containing an NF-κB binding site (*Figure 6E*). Using a chromatin immunoprecipitation (ChIP) assay, we confirmed the binding of NF-κB to a sequence located in the *Slc38a2* promoter region (*Figure 6F&G*). As expected, blockade of NF-κB activity by ammonium pyrrolidinedithiocarbamate, an NF-κB inhibitor, completely abolished hyperosmolarity-induced *Slc38a2* mRNA and protein expression (*Figure 6H–J*). Together, these findings demonstrate that the induction of SLC38A2 expression by hyperosmotic stress is dependent on the activation of the NF-κB signalling pathway.

## Overexpression of SLC38A2 protects mIMCD3 cells from hyperosmolarity-induced ferroptosis

To characterise the role of SLC38A2 in the survival of IMCD cells under hyperosmotic stress, we overexpressed SLC38A2 in mIMCD3 cells by infecting the cells with either an adenovirus expressing full-length *Slc38a2* (Ad-SLC38A2) or GFP (Ad-GFP) (*Figure 7A and B*), followed by hyperosmotic stress. Morphological examination and MTT assays showed that overexpression of SLC38A2 significantly attenuated hyperosmolarity-induced cell death (*Figure 7C and D*), suggesting that SLC38A2 protects mIMCD3 cells under hyperosmotic conditions. In addition, overexpression of SLC38A2 not only induced GPX4 expression in an isosmotic solution but also increased GPX4 expression under hyperosmotic stress (*Figure 7E–F*), indicating that SLC38A2 may exert anti-ferroptotic effects in IMCD3 cells by promoting GPX4 expression. To test this hypothesis, we treated Ad-SLC38A2-infected mIMCD3 cells with various doses of erastin and RSL3 and found that both ferroptosis inducers promoted ferroptosis in a dose-dependent manner under hyperosmolar conditions (*Figure 7G–H*). However, SLC38A2 overexpression abolished erastin-induced ferroptosis, with little effect on RSL3-elicited ferroptosis under hyperosmotic conditions (*Figure 7G–H*). Based on the capacity of SLC38A2 to transport amino acids for GSH biosynthesis, these findings suggested that SLC38A2 protects mIMCD3 cells from hyperosmolarity-induced ferroptosis, possibly by increasing intracellular GSH content and GPX4 activity.

## Inhibition and downregulation of SLC38A2 aggravate hyperosmolarity-induced ferroptosis in IMCD cells

To further confirm the anti-ferroptotic effect of SLC38A2 in IMCD cells, we treated mIMCD3 cells with 2-(methylamino) isobutyric acid (MeAIB), a competitive inhibitor of System A transporters, including SLC38A2, under isosmotic and hyperosmotic conditions. MTT assay showed that MeAIB treatment promoted cell death under both isosmotic and hyperosmotic conditions (*Figure 8A*). Conversely, knockdown of SLC38A2 expression in mIMCD3 cells (*Figure 8B*, *Figure 8—figure supplement 1*) and knockout of SLC38A2 in primary IMCD cells (*Figure 8C&D*) predisposed the cells to hyperosmolarity-induced ferroptosis. In addition, western blotting analysis showed that under hyperosmotic conditions (900 and 1200 mOsm), *Slc38a2*-gene deficient cells exhibited more robust suppression of GPX4 protein expression than WT control cells (*Figure 8E–F*), further suggesting that SLC38A2 ameliorates ferroptosis by increasing GPX4 protein expression. In support of this, Ad-SLC38A2 infection conferred cytoprotection against hyperosmolarity-induced cell death in both WT and *Slc38a2* gene-deficient primary IMCD cells (*Figure 8G*). These findings demonstrate that inhibition of SLC38A2 aggravates IMCD cell death, whereas overexpression of SLC38A2 reduces IMCD cell death under hyperosmotic conditions. In addition, the protective effect of SLC38A2 in IMCD cells was abolished by the GPX4 inhibitor RSL3 and partially attenuated by system Xc⁻ (*Figure 8H–I*). Moreover, MeAIB, an SLC38A2

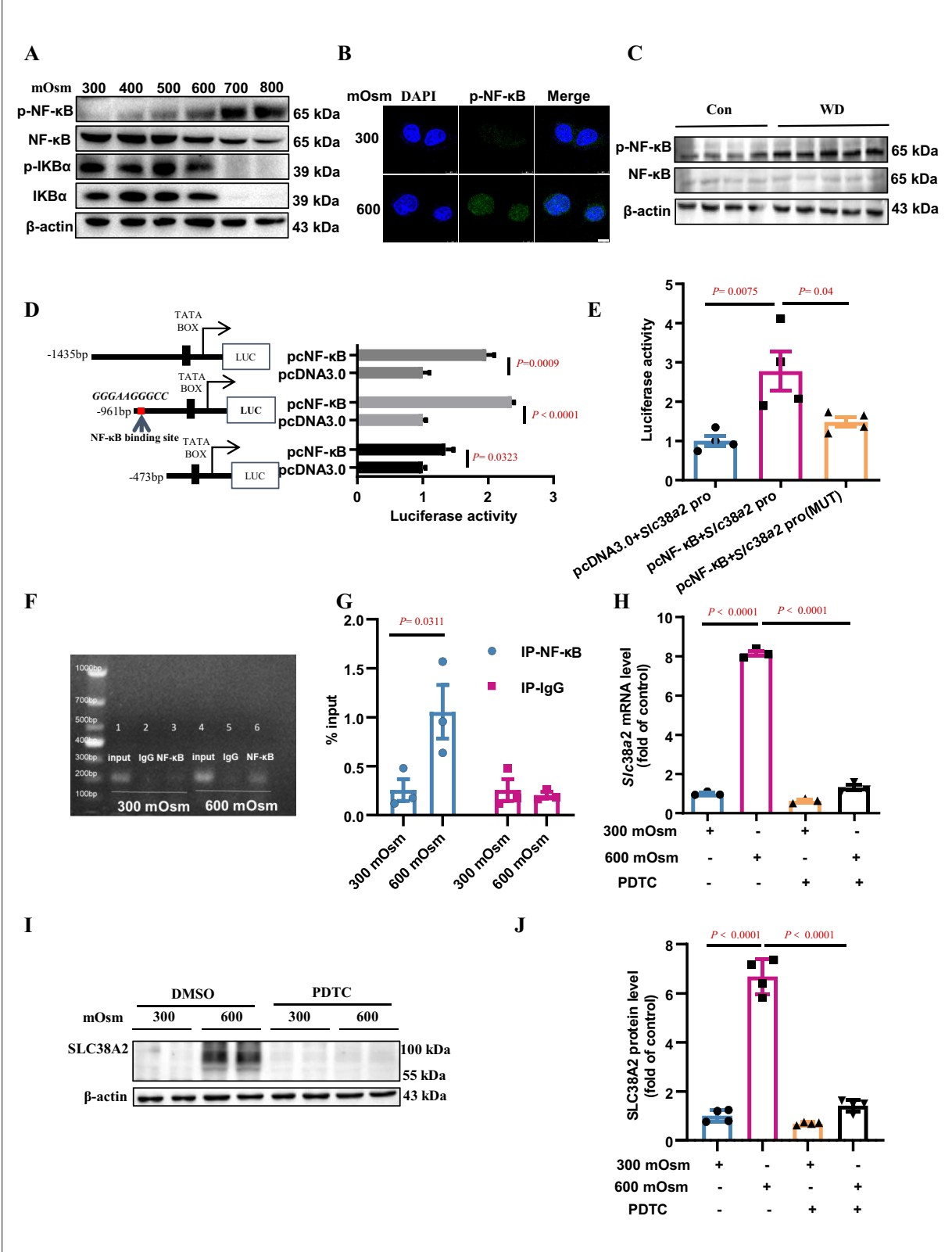

**Figure 6.** Hyperosmolarity induces SLC38A2 expression by activating the nuclear factor kappa B (NF-κB) pathway. (**A**) Hyperosmolarity increased NF-κB activity in mIMCD3 cells in a dose-dependent manner. mIMCD3 cells were treated with hyperosmotic solutions for 12 h. The protein levels of total and phosphorylated NF-κB and IκBα were measured by western blotting. (**B**) Immunofluorescence assay demonstrated that hyperosmolarity promoted nuclear translocation of NF-κB in mIMCD3 cells. The cells were exposed to hyperosmotic stress (600 mOsm) for 12 h. Scale bar = 8 μm.

*Figure 6 continued on next page*

*Figure 6 continued*

(**C**) Water deprivation significantly increased p-NF-$\kappa$B protein levels and the ratio of p-NF-$\kappa$B to total NF-$\kappa$B in the renal medulla of WT C57BL/6 mice. The mice were subjected to water deprivation for 24 h or free access to water. (**D**) Luciferase reporter assay showed that NF-$\kappa$B promoted mouse *Slc38a2* gene promoter activity. Three promoter fragments (−1435 bp~+124 bp, −961 bp~+124 bp, −473 bp~+124 bp) were obtained by PCR and subcloned into the pGL3-basic vector. n=4. (**E**) Point mutations defined a consensus sequence containing an NF-$\kappa$B binding site in mouse *Slc38a2* gene promoter. A potential NF-$\kappa$B binding site between −867 nt and −876 nt was predicted, and the sequence was mutated from 5′-GGG AAG GGC C-3′ to 5′-TCA AGT CTC C-3′. n=4. (**F and G**) Chromatin immunoprecipitation assay confirmed the binding of NF-$\kappa$B to the *Slc38a2* promoter region. n=3. (**H**) Real-time PCR analysis showed that inhibition of NF-$\kappa$B abolished hyperosmolarity-induced Slc38a2 mRNA expression. mIMCD3 cells were treated with the NF-$\kappa$B inhibitor PDTC (10 μM) for 24 h, with hyperosmotic exposure for the last 12 h. n=3. (**I and J**) Western blots showed that inhibition of NF-$\kappa$B blocked hyperosmolarity-induced SLC38A2 protein expression. n=4. Data are means ± SEM; two-tailed Student's *t*-test for D; one-way ANOVA tests for E, G, H, and J. Con, control mice with free access to water; WT, wild-type; KO, knockout; WD, water deprivation; PDTC, ammonium pyrrolidinedithiocarbamate. See numerical source data and uncropped western blot images in *Figure 6—source data 1*.

The online version of this article includes the following source data for figure 6:

**Source data 1.** Numerical and uncropped western blot source data for *Figure 6*.

inhibitor, sensitised mIMCD3 cells to erastin-induced ferroptosis under hyperosmolar conditions (*Figure 8J*). These results further support the conclusion that the protective effect of SLC38A2 against hyperosmolarity-induced cell death is largely dependent on its anti-ferroptosis action by increasing GPX4 activity.

## SLC38A2 protects IMCD cells from hyperosmotic stress-induced ferroptosis by activating mTORC1

SLC38A2-transported amino acids have been reported to activate the mTORC1 signalling pathway to promote cell growth *Cormerais et al., 2020*. To test whether mTORC1 contributes to the SLC38A2-mediated protection of IMCD cells under hyperosmotic stress, we determined the effect of hyperosmolarity on mTORC1 activity. We found that hyperosmolality suppressed the total and phosphorylated levels of rpS6 and 4EBP1, two downstream effectors of mTORC1, in mIMCD3 cells in a dose-dependent manner (*Figure 9A*, *Figure 9—figure supplement 1A*), suggesting that suppressed mTORC1 activity may be associated with reduced cell viability in IMCD cells exposed to hyperosmolarity. In support of this, inhibition of mTORC1 by rapamycin and torin-1 sensitised mIMCD3 cells to hyperosmotic stress-induced cell death in the presence of the ferroptosis inducer erastin (*Figure 9B&C*). Although rapamycin and torin-1 treatments increased GPX4 protein expression under hyperosmotic conditions (*Figure 9—figure supplement 1B and C*), mTORC1 inhibition by both compounds resulted in a significant decrease in GPx activity (*Figure 9D*) and a marked increase in cellular lipid ROS levels (*Figure 9E*). The induction of GPX4 protein expression by rapamycin and torin-1 may reflect positive feedback regulation of reduced GPX4 activity. Collectively, these results indicate that mTORC1 attenuates ferroptosis by maintaining intracellular GPX4 activity and blocking lipid peroxidation.

Furthermore, we found that adenovirus-mediated SLC38A2 overexpression completely abolished the hyperosmolarity-induced suppression of total and phosphorylated rpS6 and 4EBP1 protein levels (*Figure 9F*, *Figure 9—figure supplement 1D*), suggesting that SLC38A2 protects IMCD cells from ferroptosis by activating mTORC1. In support of this, the protective effect of SLC38A2 on hyperosmolarity-induced cell death was completely diminished by the inhibition of mTORC1 (*Figure 9G–H*). This result indicates that, in addition to system Xc⁻, SLC38A2 represents another important amino acid transporter that transports amino acids into cells to maintain the cellular glutathione level, thereby blocking ferroptosis. Collectively, our findings demonstrate that SLC38A2 protects IMCD cells from hyperosmotic stress-induced ferroptosis by activating mTORC1.

## Proposed model of SLC38A2 in the regulation of cell ferroptosis in renal MCDs under hyperosmotic stress

We present a schematic (*Figure 10*) showing the underlying mechanism by which SLC38A2 promotes the survival of renal MCD cells under hyperosmotic stress by inhibiting ferroptosis. (1) Hyperosmotic stress increases lipid ROS production, which promotes ferroptosis of MCD cells; (2) hyperosmolarity upregulates *Slc38a2* mRNA expression by activating the NF-κB pathway; (3) SLC38A2 mediates the uptake of Cys, Gly, and Gln, which may help maintain cellular redox homeostasis and intracellular GPX4 activity to exert its anti-ferroptotic effect; (4) SLC38A2-mediated uptake of glutamine can

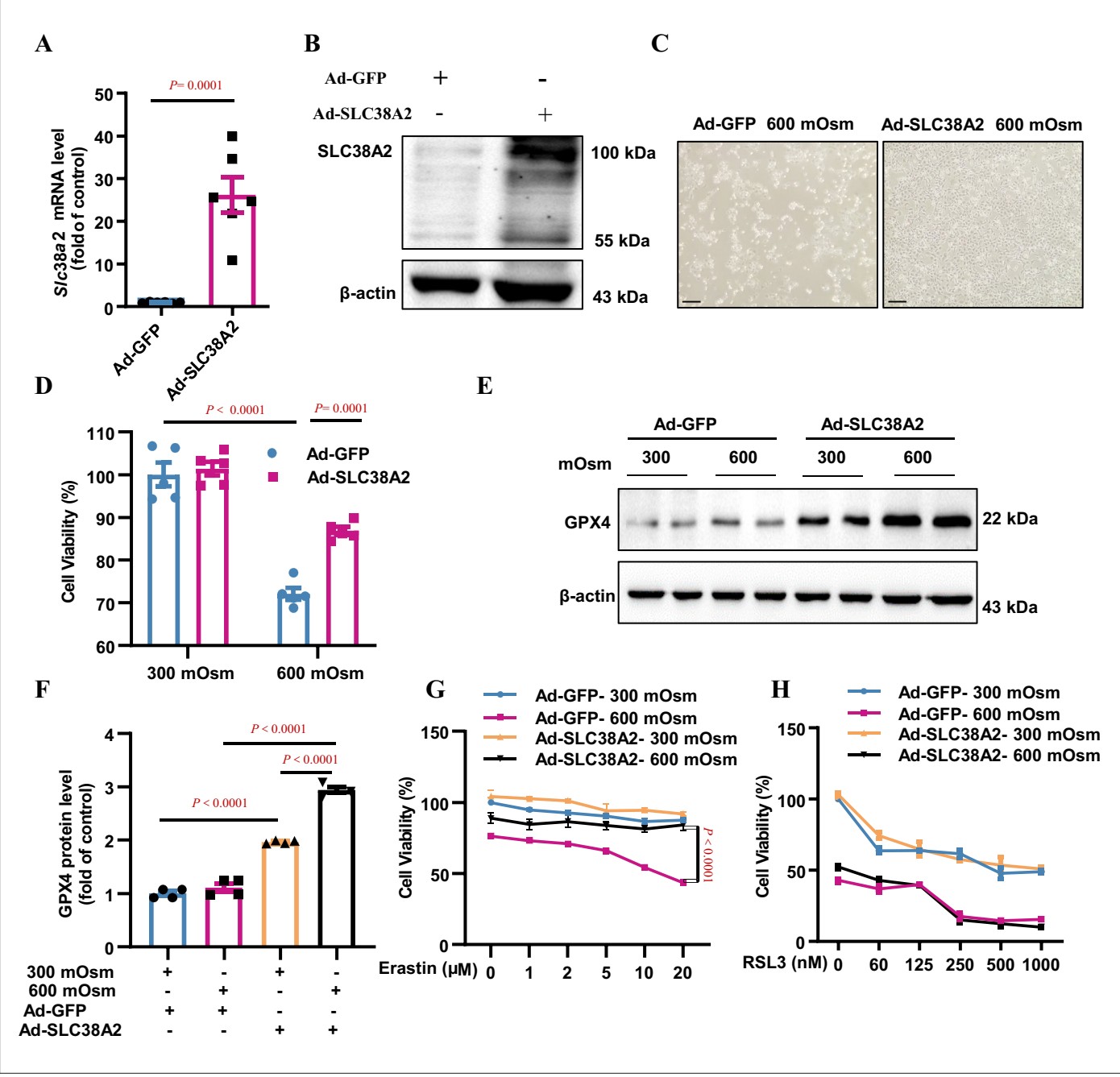

**Figure 7.** Overexpression of SLC38A2 protects mIMCD3 cells from hyperosmolarity-induced ferroptosis. (**A and B**) Adenovirus-mediated SLC38A2 overexpression at both the mRNA (**A**) and protein (**B**) levels. mIMCD3 cells were infected with the Ad-SLC38A2 or Ad-GFP (multiplicity of infection [MOI] = 20) for 24 h. Real-time PCR and western blot assays showed that Ad-SLC38A2 infection significantly increased *Slc38a2* mRNA (**A**) and protein (**B**) expression. n=6. (**C**) Morphological examination showing the protective effect of SLC38A2 overexpression on hyperosmolarity-induced mIMCD3 cell death. The cells were infected with Ad-SLC38A2 or Ad-GFP for 36 h (MOI = 20), followed by exposure to hyperosmotic stress (600 mOsm) for 12 h. Scale bar = 100 μm. (**D**) MTT assay demonstrating that SLC38A2 overexpression protects mIMCD3 cells from hyperosmolarity-induced cell death. n=5. (**E and F**) Western blotting analysis results showing that SLC38A2 overexpression increased GPX4 protein expression under both isosmotic and hyperosmotic conditions. The cells were treated as described in (**C**). n=4. (**G**) SLC38A2 protects erastin-induced ferroptosis of mIMCD3 cells under hyperosmotic stress. The cells were infected with Ad-SLC38A2 or Ad-GFP (MOI = 20) for 36 h, followed by treatment with erastin at various doses under hyperosmotic stress (600 mOsm) for 12 h. n=8. (**H**) The GPX4 inhibitor, RSL3, abolished the protective effect of SLC38A2 overexpression in hyperosmolarity-treated mIMCD3 cells. The cells were treated as described in (**G**). MTT assay was used to determine cell viability. RSL3 reduced cell viability in the hyperosmotic group to a greater extent than in the isosmotic group. n=3. Data are means ± SEM; two-tailed Student's *t*-test for A; one-way ANOVA tests for D and F; two-way ANOVA tests for **G** and **H**. See numerical source data and uncropped western blot images in *Figure 7—source data 1*.

*Figure 7 continued on next page*

*Figure 7 continued*

The online version of this article includes the following source data for figure 7:

**Source data 1.** Numerical and uncropped western blot source data for *Figure 7*.

strongly activate mTORC1 by facilitating cellular uptake of leucine via the System L amino acid transporter LAT1; (5) SLC38A2 promotes Gln uptake, which is converted into α-ketoglutarate to activate mTORC1. Note: Erastin treatment decreases System Xc⁻, resulting in the reduction of Cys uptake and depletion of GSH, leading to a decrease in GPX4 enzymatic activity and an increase in lipid peroxidation. RSL3 blocks GPX4 activity, which decreases the cellular antioxidant capacity, leading to the accumulation of lipid ROS and ferroptosis. Liproxstatin-1 (Lip-1) and ferrostatin-1 (Fer-1) exert anti-ferroptotic effects by attenuating lipid peroxidation.

## Discussion

The hyperosmotic state of the renal medulla is essential for urine concentration. However, the underlying mechanism by which renal MCD cells survive harsh hyperosmotic stress remains incompletely understood. The present study demonstrated a critical role for the sodium-dependent neutral amino acid transporter SLC38A2 in the survival of MCD cells under hyperosmolar conditions. First, we clarified that ferroptosis is a major type of RCD in IMCD cells under hyperosmotic stress. Second, we identified SLC38A2 as a previously uncharacterised osmoresponsive amino acid transporter in the kidney, where its expression is significantly induced by hyperosmolality. Third, we provided a compelling evidence that SLC38A2 plays an important role in protecting IMCD cells from hyperosmotic stress-induced ferroptosis, mainly by activating the mTORC1 signalling pathway. Finally, using *Slc38a2-* knockout mice, we confirmed the critical role of SLC38A2 in maintaining renal medullary homeostasis and renal urine concentration by attenuating medullary cell ferroptosis. Collectively, our findings demonstrate that *Slc38a2* is a critical osmoprotective gene in the renal medulla, and its appropriate function is a prerequisite for urine concentration.

MCD cells are essential for urine concentration and must adapt to constant changes in environmental osmolarity to maintain their viability and functionality (*Woo and Kwon, 2002*; *Dmitrieva and Burg, 2005*). However, the underlying mechanism by which MCD cells survive in harsh hyperosmotic environments remains unclear. In the past decade, it has been repeatedly reported that apoptosis is involved in hyperosmolarity-induced cell death in MCD cells, particularly IMCD cells, and multiple factors including TonEBP, NF-κB, P53, PPARβ/δ, and FXR protect MCD cell viability through their anti-apoptotic properties (*Han et al., 2011*; *Xu et al., 2018*; *Guan, 2004*; *Dmitrieva et al., 2001*). However, emerging evidence suggests that in addition to apoptosis, other types of RCD may also contribute to hyperosmolarity-elicited death of MCD cells, as apoptotic cells only account for a small portion of the injured cell population (*Xu et al., 2018*; *Zhang et al., 2014*). The present study provides a compelling evidence that ferroptosis represents the major type of RCD in hyperosmolarity-induced cell death, as the ferroptosis inhibitors, liproxstatin-1 and ferrostatin-1, almost completely abolish, whereas the ferroptosis inducers, erastin and RSL-3, significantly increase, hyperosmotic stress-induced injury of MCD cells.

The present study identified *Slc38a2* as an osmoresponsive gene in MCD cells. RNA-Seq analysis revealed 9266 genes differentially expressed in hyperosmolarity-treated mIMCD3 cells, including many previously reported osmoregulatory genes, such as *Hspa1a*, *Hspa1b*, *Hspa41*, *Nfat5* (*TonEBP*), and *Hif1α* (*Yang et al., 2005*; *Woo et al., 2002*). In addition, an array of genes involved in amino acid transport was induced by hyperosmolar treatment. Among the amino acid transporters expressed in the mouse kidney, SLC38A2 is the major transporter in the SLC38 subfamily, with high abundance in renal MCD cells, where its expression is significantly induced by hyperosmolarity. In the present study, we also provide clear evidence that SLC38A2 is directly regulated by NF-κB, which can drive transcription of the *Slc38a2* gene by binding to its gene promoter. Therefore, *Slc38a2* is a novel osmoresponsive gene in IMCD cells, and its expression is under the direct control of NF-κB during hyperosmotic conditions.

SLC38A2 can transport a group of neutral amino acids, including cysteine, glutamine, glycine, alanine, asparagine, histidine, methionine, proline, and serine (*Mackenzie and Erickson, 2004*; *Shen et al., 2022*). It has been previously reported that an increase in intracellular glutamine, a common

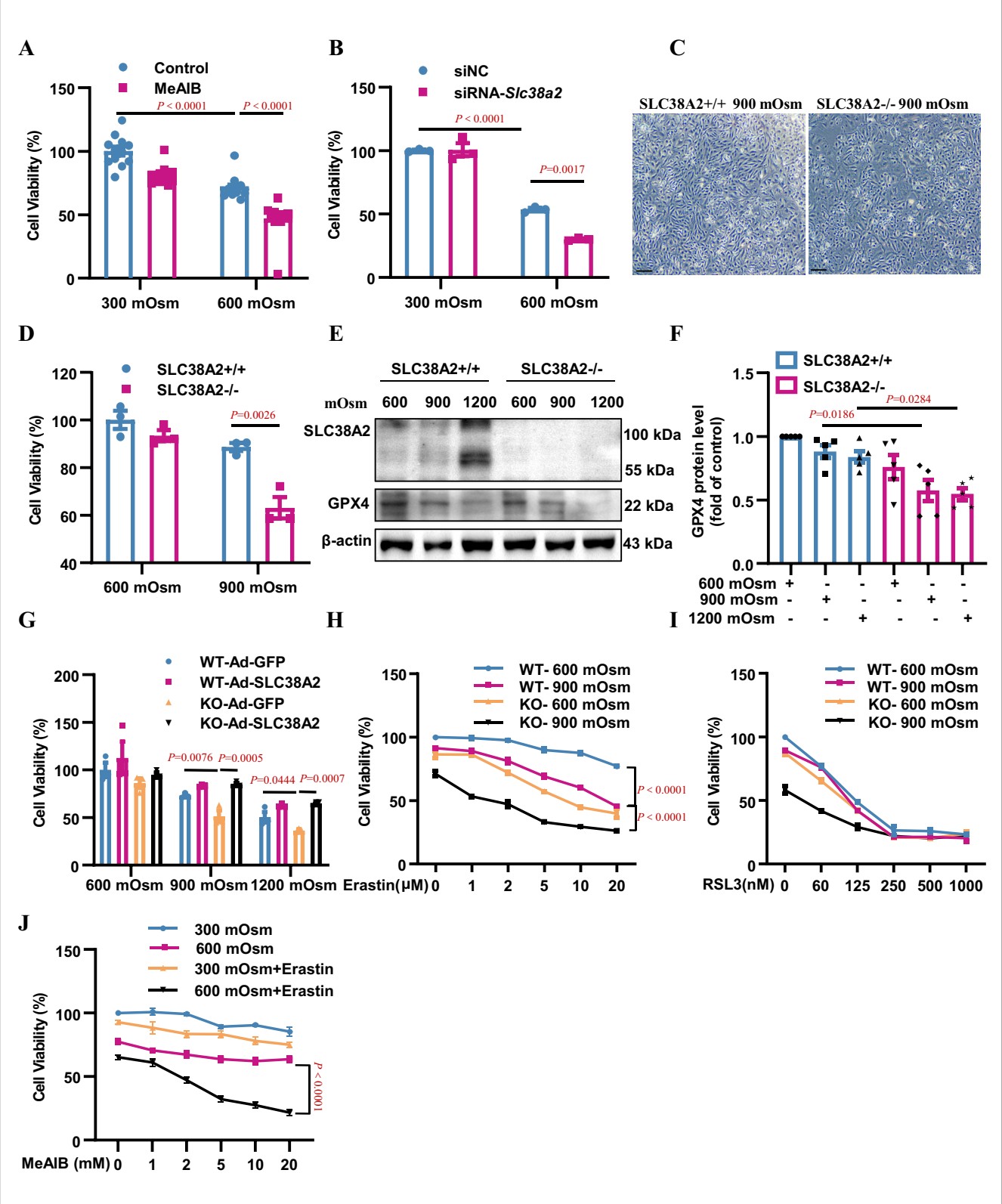

**Figure 8.** Inhibition or downregulation of SLC38A2 aggravates hyperosmolarity-induced ferroptosis in mIMCD3 and primary medullary collecting duct (MCD) cells. (**A**) The System A amino acid transporter inhibitor 2-(methylamino) isobutyric acid (MeAIB) worsened hyperosmolarity-induced cell death. mIMCD3 cells were treated with 5 mM MeAIB for 24 h, with or without hyperosmotic exposure (600 mOsm) in the final 12 h. MTT assay was used to determine cell viability. n=12. (**B**) SLC38A2 knockdown sensitised mIMCD3 cells to hyperosmolarity-elicited cell death. The cells were transfected with

*Figure 8 continued on next page*

*Figure 8 continued*

*Slc38a2* small interfering RNA (siRNA) for 24 h, followed by exposure to hyperosmotic stress (600 mOsm) for 12 h. n=3. (**C**) *Slc38a2*-gene deficiency promoted hyperosmolarity-induced cell death. The primary inner MCD (IMCD) cells of the wild-type (*Slc38a2*⁺/⁺, WT) and knockout (*Slc38a2*⁻/⁻, KO) mice were challenged with hyperosmotic stress (900 mOsm) for 12 h. The cell numbers were examined by light microscopy. Scale bar = 100 μm. (**D**) MTT assay of primary MCD cells of WT and KO mice. The cells were treated with hypertonicity (900 mOsm) for 12 h. n=3. (**E and F**) Western blotting analysis results showing that hyperosmolarity caused a more significant decrease in GPX4 levels in SLC38A2⁻/⁻ IMCD cells than in SLC38A2⁺/⁺ IMCD cells under hyperosmotic conditions. n=5. (**G**) MTT experiment results demonstrating that adenovirus-mediated SLC38A2 overexpression completely reversed hyperosmotic stress-induced cell injury of primary SLC38A2⁻/⁻ IMCD cells. n=3. (**H**) The System Xc⁻ inhibitor erastin sensitised hyperosmolality-induced ferroptosis in SLC38A2⁻/⁻ IMCD cells in a dose-dependent manner. Erastin also enhanced hyperosmotic stress-elicited ferroptosis in SLC38A2⁺/⁺ IMCD cells to a lower extent than that in SLC38A2⁻/⁻ MCD cells. n=3. (**I**) The GXP4 inhibitor RSL3 abolished the difference in cell viability under hyperosmotic stress between primary SLC38A2⁺/⁺ and SLC38A2⁻/⁻ IMCD cells in a dose-dependent manner. n=3. (**J**) MTT assay results demonstrating that the SLC38A2 blocker MeAIB sensitised mIMCD3 cells to erastin-induced ferroptosis under hyperosmolarity. mIMCD3 cells were treated with MeAIB at various doses for 24 h, with or without erastin (10 μM) treatment in the presence or absence of hyperosmolarity (600 mOsm) for the last 12 h. n=4. siNC, scramble siRNA control. Data are means ± SEM; one-way ANOVA tests for A, B, D, F, and G; two-way ANOVA tests for H, I, and J. See numerical source data and uncropped western blot images in *Figure 8—source data 1*.

The online version of this article includes the following source data and figure supplement(s) for figure 8:

**Source data 1.** Numerical and uncropped western blot source data for *Figure 8*.

**Figure supplement 1.** Knockdown of *Slc38a2* expression in mIMCD3 cells using a small interfering RNA (siRNA)-mediated approach.

**Figure supplement 1—source data 1.** Numerical and uncropped western blot source data for *Figure 8—figure supplement 1*.

transport substrate for SLC38A2, is related to cell volume recovery under hyperosmotic conditions (*Franchi-Gazzola et al., 2006*). During hyperosmotic conditions, System A transport activity increases in renal epithelial cells, tissue fibroblasts, vascular smooth muscle cells, lymphocytes, and endothelial cells (*Franchi-Gazzola et al., 2006*; *Chen and Kempson, 1995*; *Yamauchi et al., 1994*; *Trama et al., 2002*; *Petronini et al., 2000*). The application of *Slc38a2* small interfering RNA (siRNA) to inhibit its expression can hinder the increase in intracellular amino acid content and delay the recovery of cell volume (*Bevilacqua et al., 2005*). Therefore, SLC38A2 plays an important role in the regulation of cell volume upon hyperosmotic exposure, and the amino acids transported by SLC38A2 act as organic osmolytes in hyperosmotically stressed cells (*Franchi-Gazzola et al., 2006*). In addition, emerging evidence shows that SLC38A2 can also transport betaine (*Nishimura et al., 2014*), an important osmolyte contributing to the osmoprotection of the MCD cells of dehydrated animals. Therefore, the induction of SLC38A2 in hyperosmotically stressed cells and in the renal medulla of dehydrated mice may help restore cell volume and maintain cell viability and function of MCD cells.

In the present study, we report, for the first time, a critical role of SLC38A2 in protecting IMCD cells against hyperosmolarity-induced ferroptosis by increasing cellular antioxidant capacity. Unrestrained lipid peroxidation is a hallmark of ferroptosis (*Chen et al., 2021*). The reduction of lipid peroxidation is essential for cell survival and requires functional GPX4 and GSH (*Jiang et al., 2021*). Cys is the rate-limiting substrate in the biosynthesis of reduced GSH. Intracellular Cys and GSH levels are highly dependent on the system Xc⁻ cystine/glutamate antiporter (*Yang et al., 2022*). In addition, Glu, which can be converted from Gln and Met and metabolised into Cys and Gly, is also important for producing GSH to reduce lipid peroxidation via GPX4 (*Hensley et al., 2013*). Therefore, SLC38A2, as a Cys, Gln, Met, and Gly transporter whose expression is abundantly present in IMCD cells and significantly induced by hyperosmolarity, may help maintain cellular redox homeostasis and intracellular GPX4 activity to exert its anti-ferroptotic effect in IMCD cells both in vitro and in vivo (*Figure 10*). In addition, we found that hyperosmotic stress reduced GPX4 levels in cultured IMCD cells, which could be restored by SLC38A2 overexpression. Conversely, water deprivation downregulated GPX4 expression in the renal medulla of WT mice, which was further reduced in *Slc38a2*⁻/⁻ mice. These findings demonstrate that SLC38A2 may protect MCD cells from ferroptosis by increasing GPX4 expression and activity.

In addition to supplying sufficient Cys, Glu, Met, and Gly for GSH biosynthesis in IMCD cells under hyperosmotic stress, SLC38A2 may also exert its anti-ferroptotic effect by activating the mTORC1 signalling pathway. mTORC1 integrates multiple signal inputs, including energy, growth factors, and amino acids, to promote cell growth and proliferation by phosphorylating S6K1 and 4EBP1 (*Zhang et al., 2021*). In the present study, we found that hyperosmolarity suppressed IMCD cell viability by inhibiting mTORC1 activity, which was markedly attenuated by SLC38A2 overexpression. It has been

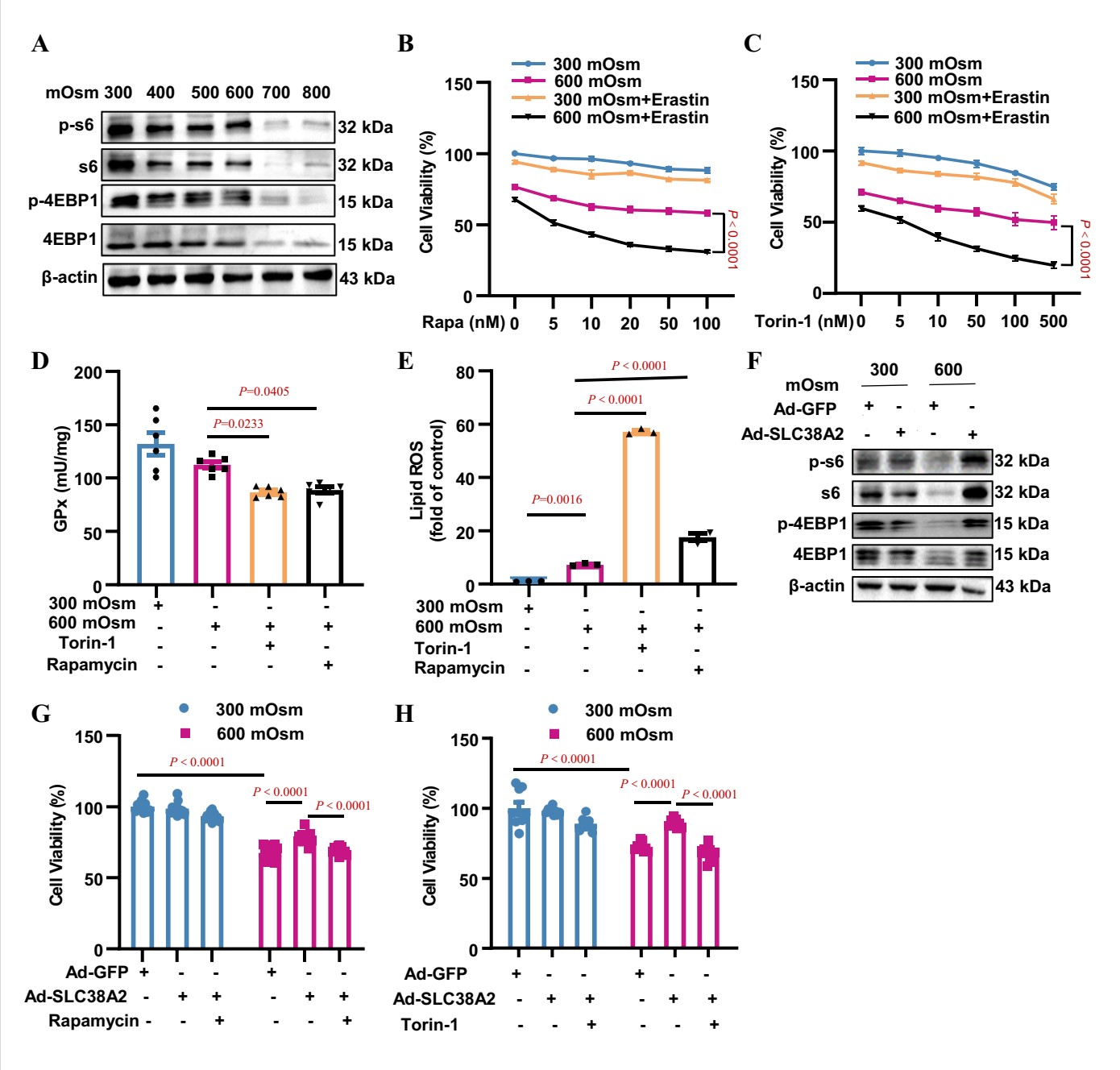

**Figure 9.** SLC38A2 protects mIMCD3 cells from ferroptosis under hyperosmotic stress by activating mTORC1. (**A**) Inhibitory effect of hyperosmolarity on the mTORC1 signalling pathway. mIMCD3 cells were incubated with hyperosmotic solutions for 12 h. Western blotting analysis showed that hyperosmolarity suppressed the total and phosphorylated S6 and 4E-BP1 expression in a dose-dependent manner. (**B and C**) MTT assay results showing that rapamycin (**B**) or torin-1 (**C**) sensitised mIMCD3 cells to erastin-induced ferroptosis under hyperosmolality. mIMCD3 cells were treated with rapamycin or torin-1 at various doses for 24 h. During the last 12 h, the cells were treated with or without erastin (10 μM) in the presence or absence of hyperosmolarity (600 mOsm). n=8. (**D**) The mTORC1 inhibitors, rapamycin and torin-1, reduced the activity of glutathione peroxidase (GPx) activity. The cells were treated with rapamycin (10 nM) or torin-1 (10 nM) for 24 h, with or without hyperosmotic treatment in the final 12 h. n=6. (**E**) Cytometry assay showing that rapamycin and torin-1 significantly increased the levels of lipid reactive oxygen species (ROS). n=3. (**F**) Western blotting assay results demonstrating that SLC38A2 overexpression reversed hyperosmolarity-suppressed expression of total and phosphorylated S6 and 4E-BP1. The cells were infected with Ad-SLC38A2 or Ad-GFP for 36 h followed by hyperosmotic stress for 12 h. (**G–H**) MTT assay results showing that rapamycin (**G**) and torin-1 (**H**) completely abolished the protective effect of SLC38A2 in mIMCD3 cells under hyperosmolarity. After being infected with the adenoviruses, the cells were treated with rapamycin (10 nM) or torin-1 (10 nM) for 24 h with or without hyperosmolarity in the last 12 h. n=9–12. Data are means ± SEM;

*Figure 9 continued on next page*

*Figure 9 continued*

two-way ANOVA tests for B and C; one-way ANOVA tests for D, E, G, and H. See numerical source data and uncropped western blot images in *Figure 9—source data 1*.

The online version of this article includes the following source data and figure supplement(s) for figure 9:

**Source data 1.** Numerical and uncropped western blot source data for *Figure 9*.

**Figure supplement 1.** SLC38A2 protects mIMCD3 cells from ferroptosis under hyperosmotic stress by activating mTORC1.

**Figure supplement 1—source data 1.** Numerical and uncropped western blot source data for *Figure 9—figure supplement 1*.

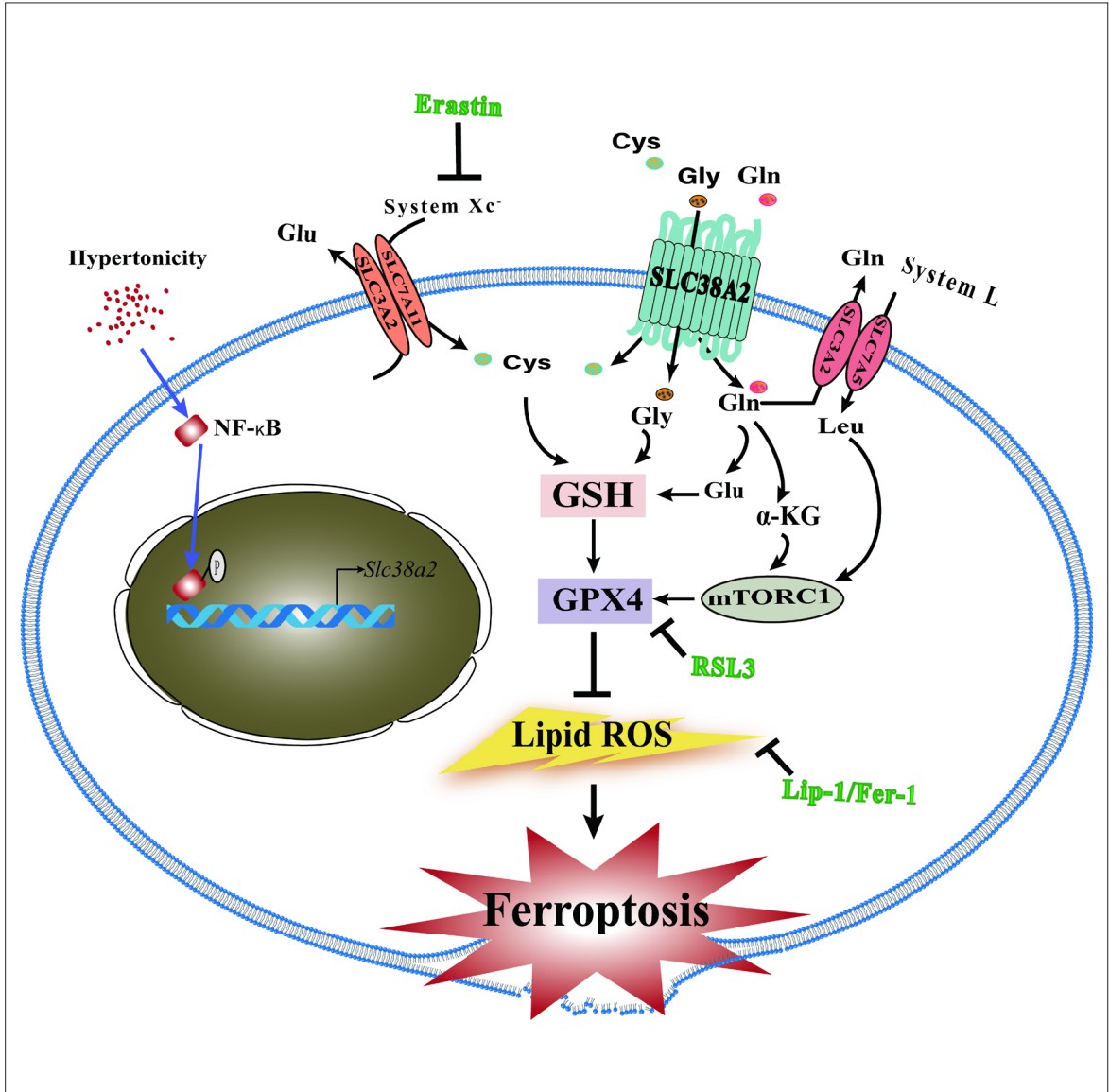

**Figure 10.** Schematic showing the underlying mechanism by which SLC38A2 promotes the survival of renal medullary collecting duct (MCD) cells under hyperosmotic stress by inhibiting ferroptosis. (1) Hyperosmotic stress increases lipid reactive oxygen species (ROS) production, which promotes ferroptosis of MCD cells; (2) hyperosmolarity upregulates *Slc38a2* mRNA expression by activating the nuclear factor kappa B (NF-$\kappa$B) pathway; (3) SLC38A2 mediates the uptake of Cys, Gly, and Gln, which may help maintain cellular redox homeostasis and intracellular glutathione peroxidase 4 (GPX4) activity to exert its anti-ferroptotic effect; (4) SLC38A2-mediated uptake of glutamine can strongly activate mTORC1 by facilitating cellular uptake of leucine via the System L amino acid transporter LAT1; (5) SLC38A2 promotes Gln uptake, which is converted into α-ketoglutarate (α-KG) to activate mTORC1.

previously reported that glutamine, a well-documented SLC38A2 substrate, can strongly activate mTORC1 by facilitating cellular uptake of leucine via the System L amino acid transporter SLC7A5/SLC3A2 (*Nicklin et al., 2009*; *Cohen and Hall, 2009*; *Figure 10*). Therefore, it is reasonable to draw a conclusion that hyperosmolarity-induced SLC38A2 expression may protect the MCDs of dehydrated mice through increasing mTORC1 expression and activity.

In the present study, we found that the *Slc38a2*-knockout mice exhibited a slight increase in both urine output and osmolyte excretion. It is speculated that as an amino acid transporter, *Slc38a2*-gene deficiency results in blunted cellular uptake of certain amino acids by peripheral tissues and elevated levels of amino acids in the plasma, which cause mild osmotic diuresis. Under dehydrated conditions, enhanced collecting duct ferroptosis may also contribute to increased urine output in the *Slc38a2*-knockout mice. In addition, elevated circulating amino acid levels and increased glomerular filtration of amino acids into the renal tubules may be responsible for a slight increase in urinary osmolarity observed in the *Slc38a2*-knockout mice.

Taken together, our data demonstrate that (1) ferroptosis is the major type of RCD in hyperosmolarity-exposed IMCD cells; (2) *Slc38a2* is a novel osmoresponsive gene in IMCD cells, and its expression is significantly induced by NF-κB under hyperosmotic stress; (3) SLC38A2 protects IMCD cells from hyperosmolality-induced ferroptosis both in vitro and in vivo; and (4) the renoprotective effect of SLC38A2 is mediated by the activation of the mTORC1 signalling pathway. Altogether, our study demonstrates that SLC38A2-mediated anti-ferroptotic effects represent a novel and critical mechanism by which renal MCD cells can survive and function in a harsh hyperosmotic environment. Pharmacological blockade of the SLC38A2-mTORC1 axis by the SLC38A2 inhibitor and rapamycin may disrupt renal medullary architecture and impair urine concentration by promoting ferroptosis of MCD cells.

## Materials and methods

### Animal studies

Male C57BL/6 WT mice were purchased from the Vital River Laboratory Animal Technology (Beijing, China). Global *Slc38a2*-knockout (*Slc38a2$^{-/-}$*) mice were generated on a C57BL/6 background by deleting a 10-bp sequence (5′-CCA CAA TCG C-3′) in exon 4 using the CRISPR/Cas9 technique. Owing to the high incidence of postnatal lethality in *Slc38a2*-knockout mice (*Slc38a2$^{-/-}$*) on a C57BL/6 background (*Weidenfeld et al., 2021*), we crossed *Slc38a2* heterozygous mice with WT mice on a 129 background for five generations to obtain developmentally normal *Slc38a2*-knockout mice for in vivo studies. The genotype was confirmed by DNA sequencing (*Figure 4—figure supplement 1*). Male mice aged 8–10 weeks were used in this study. All animals had free access to food and were either allowed free access to water or subjected to water restriction for 24 h. After water restriction, the mice were sacrificed and perfused with cold PBS. The left kidneys were fixed and paraffin-embedded, whereas the right kidneys were used to dissect the renal cortex and medulla.

### Chemicals and reagents

α-MeAIB was purchased from Sigma-Aldrich (Cat#M2383). Erastin (Cat#B1524), (1 S,3R)-RSL3 (Cat#B6095), liproxstatin-1 (Cat#B4987), and ferrostatin-1 (Cat#A4371) were purchased from Applied Biosystems. Torin-1 (Cat#4858) and rapamycin (Cat#9904) were purchased from Cell Signalling Technology. Antibodies against NF-κB (Cat#8242), p-NF-κB (Ser$^{536}$, Cat#3033), IκBα (Cat#9242), p-IκBα (Ser (*Zhang et al., 2014*), Cat#2859), rpS6 (Cat#2217), p-S6 (Ser$^{235/236}$, Cat#4858), 4E-BP1 (Cat#9644), p-4E-BP1 (Thr$^{37/46}$, Cat#2855), GPX4 (Cat#52455), caspase-3 (Cat#9662), and cleaved caspase-3 (Cat#9664) were purchased from Cell Signalling Technology. Antibodies against 4-HNE (Cat#MAB3249) and 8-oxo-dG (Cat#4354-MC-050) were purchased from R&D Systems (Minneapolis, MN, USA). Primary antibodies against AQP1 (Cat#sc25287) and THP (Cat#sc271022) were purchased from Santa Cruz Biotechnology, and antibodies against NCC (Cat#ab95302) and AQP2 (Cat#ab199975) were purchased from Abcam. Primary antibodies against β-actin (Cat. #60008–1-Ig) were purchased from Proteintech. An antibody against SLC38A2 for western blotting was purchased from MBL (Cat#BMP081), and an antibody against SLC38A2 for immunohistochemical and immunofluorescence staining was purchased from NOVUS (Cat#NBP1-88872).

## Culture of mIMCD3 cell line

The established mIMCD3 murine IMCD cell line was originally developed by Rauchman and provided by the American Type Culture Collection. mIMCD3 cells were cultured in Dulbecco's Modified Eagle Medium/Nutrient Mixture F-12 ( DMEM/F12 ) medium containing 10 mM HEPES, 2 mM L-glutamine, 100 IU/mL penicillin, and 100 IU/mL streptomycin at 37°C in a humidified 5% $CO_2$ atmosphere. For mIMCD3 cells, short tandem repeats identification was performed, and the results showed that the cells were free of mycoplasma contamination.

## Primary culture of mouse inner medullary collecting duct cells

Mouse primary collecting duct cells were cultured as previously described, with minor modifications (*Xu et al., 2018*). In brief, four 8-wk-old male C57BL/6 mice were sacrificed by cervical dislocation, and the kidneys were removed under sterile conditions, placed in sterile PBS buffer, and washed three times. The kidneys were then placed in DMEM/F12 culture solution to separate the renal cortex and the medulla. Renal inner medulla samples were digested in a hyperosmotic enzyme solution containing 12 mL of DMEM/F12 supplemented with 120 mM NaCl, 80 mM urea, 24 mg collagenase (Sigma), and 8.5 mg hyaluronidase (Sigma) at 37°C for 1 h. The digested cell suspension was centrifuged at 1000 rpm for 5 min, and the supernatant was discarded. The IMCD cell pellets were finally resuspended in hyperosmotic DMEM/F12 medium containing 10 mM HEPES, 2 mM L-glutamine, 100 IU/mL penicillin, 100 IU/mL streptomycin, 50 nM hydrocortisone, 5 pM 3,3,5-triiodo-L-thyronine, 1 nM sodium selenate, 5 mg/L transferrin, and 10% foetal bovine serum. To maintain the physiological function of primary IMCD cells, the cells were grown in transwell chambers with 0.4 μm pores (Corning) and routinely maintained in a hyperosmotic (600 mOsm) medium.

## Cell ultrastructure examination by transmission electron microscopy

mIMCD3 cells were scraped using a cell scraper and centrifuged at 400 ×*g* for 10 min. After the supernatant was discarded, 3% glutaraldehyde fixative solution was added to the tube to suspend the cells. Next, the cells were fixed with 1% glutaraldehyde in 0.1 M PBS (pH 7.4) at 4°C for 2.5 h, dehydrated through an ethanol gradient, and finally embedded in Spurr's resin. The fixed cells were photographed using a transmission electron microscope (HT7800; Hitachi, Japan) at different magnifications.

## RNA-Seq analysis of the IMCD transcriptome

mIMCD3 cells were cultured in either isosmotic or hyperosmotic medium for 6 h. Total RNA was then isolated, and only samples with an RNA integrity number greater than 8.0 were selected for further analysis. After the samples were subjected to quality control, the mRNA was enriched. Sequencing and analysis were performed by Novogene Corporation (Beijing, China), and the protocol included quality control analysis, read mapping to the reference genome, transcriptome assembly, coding potential analysis, conservative analysis, target gene prediction, gene expression level quantification, and differential expression analysis. To ensure that the gene expression levels estimated from the different genes and experiments were comparable, the expected number of Fragments Per Kilobase of transcript sequence per million base pairs sequenced was used. The URL of the dataset is https://www.ncbi.nlm.nih.gov/geo/query/acc.cgi?acc=GSE206476.

## Real-time PCR

Real-time PCR was used to measure the mRNA levels of messenger RNA. Briefly, total RNA was extracted using TRIzol reagent (Thermo Fisher Scientific, Waltham, MA, USA), quantified using Nano-Drop 2000 (Thermo Fisher Scientific), and reverse-transcribed to cDNA. Primers were obtained from Thermo Fisher Scientific, according to the manufacturer's instructions, and are listed in *Supplementary file 1*. β-actin was used as an internal standard. SYBR Green (Invitrogen; Thermo Fisher Scientific) was used as the fluorochrome according to the manufacturer's instructions. PCR amplification was performed on an ABI 7300 plus (Thermo Fisher Scientific) system at 94°C for 5 min and 35 cycles of 94°C for 30 s, 59°C for 30 s, and 72°C for 30 s, followed by extension at 72°C for 5 min.

## Western blotting analysis

Mouse primary IMCD cells, mIMCD3 cells, or mouse renal medulla tissues were lysed in Radio Immunoprecipitation Assay ( RIPA ) buffer containing protease inhibitor cocktail (HY-K0010, MedChemExpress).

The lysates were then centrifuged at 12,000 ×g at 4°C for 15 min, and the supernatants were collected for immunoblotting analysis. Protein concentrations were determined using the Pierce BCA kit (Thermo Fisher Scientific). Cell lysates were mixed with SDS-PAGE loading buffer, fractionated with 10% SDS-PAGE, and transferred onto polyvinylidene fluoride membranes. The membranes were then incubated with the primary antibodies overnight at 4°C, followed by incubation with the appropriate secondary antibodies for 1 h at 25°C. Finally, the membranes were incubated with SuperLumia ECL Plus HRP Substrate Reagent, and signals from immunoreactive bands were visualised using a Chemi-luminescent Imaging System (Tanon 5200; Shanghai, China).

## Immunohistochemical assay

For immunohistochemical analysis, tissue sections (4 µm) were dewaxed and rehydrated. The slides were then immersed in 3% $H_2O_2$ for 8 min at 25°C to eliminate endogenous peroxidases. The primary antibodies were added at an appropriate dilution, followed by incubation overnight at 4°C in a wet box. The secondary antibodies labelled with the corresponding horseradish peroxidase were then applied and incubated at 25°C or 1 h. Finally, the slides were stained with 3,3'-diaminobenzidine and haematoxylin.

## Immunofluorescence assay

Mouse primary IMCD cells or mIMCD3 cells were washed three times with 1× PBS and then embedded in 4% paraffin wax (PFA) for 15 min. After the PFA was removed, the cells were incubated with 0.1% bovine serum albumin (BSA) at 25°C or 30 min. After incubation with a primary antibody against SLC38A2 or NF-κB at an appropriate dilution, the sections were treated with Fluorescein Isothiocyanate (FITC) -labelled Alexa Fluor-488- and/or Alexa Fluor-594-conjugated secondary antibody at a 1:1000 dilution for 1 h. The sections were then covered with 4',6-diamidino-2-phenylindole (DAPI), Fluoromount-G (Southern Biotech). The stained cells were visualised and photographed using a confocal microscope (Leica, Germany). DAPI and DID (C1995S; Beyotime, China) were used to visualise nuclei and cell membranes, respectively.

## TUNEL assay

The TUNEL assay was performed using the in situ cell death detection kit (KGA7073, KeyGEN BioTECH, China) following the protocol described. Briefly, the paraffin slides of the kidney were deparaffinised and rehydrated using xylene and a graded series of ethanol. Proteinase K was added and incubated for 30 min at 37°C. Drop 50 µl TdT enzyme reaction solution on slides and put them into a warm box for 60 min at 37°C, away from light. Then add 50 µL streptavidin fluorescein labelling solution to each sample, put them into a wet box, and react in a dark place at 37°C for 30 min. The nucleus was stained with DAPI staining solution, and the reaction was kept away from light at 25°C for 10 min.

## Luciferase reporter assay

A mouse *Slc38a2* promoter-driven luciferase reporter was constructed. Briefly, the mouse Slc38a2 promoter region was amplified from the tail-derived genomic DNA of a C57BL/6 mouse. The mouse *Slc38a2* promoter region containing the fragment −1799 to + 124 bp was amplified by PCR with the primers 5'-TCC TGC AGT ATG AAC CAT GGA A-3' (forward primer) and 5'- ACA GGC AGA AGA GGT GGA-3' (reverse primer). The amplified fragment was cloned into the luciferase reporter gene vector PGL3-basic (Promega, Madison, WI, USA), and the resultant construct, designated as m*Slc38a2*-luci, was sequenced to validate its orientation and sequence. Next, m*Slc38a2*-luci was truncated into three different lengths (AuGCT Biotechnology, Beijing, China), including −1435 bp to + 124 bp, −961 bp to +124 bp, and −473 bp to + 124 bp (*Figure 6D*). All constructs were confirmed by DNA sequencing. The binding sites of the transcription factors in the mouse *Slc38a2* promoter region were predicted using PROMO (http://alggen.lsi.upc.es/cgi-bin/promo_v3/promo/promoinit.cgi?dirDB=TF_8.3/). To determine luciferase activity, HEK293T cells were seeded in 24-well plates at a density of 2.0 × $10^5$ cells/well. After grown to 50–60% confluence, the cells were transfected with the plasmids using the Lipofectamine 3000 Transfection Reagent. Mouse NF-κB expression plasmid (250 ng; OriGene Technologies, Beijing, China), *Slc38a2* gene promoter-driven luciferase reporter construct (250 ng), and an internal reference PRL-CMV plasmid (25 ng; Promega) were transfected into HEK293T cells.

After incubation for 24 h, the cells were harvested in luciferase lysis buffer for the detection of luciferase activity, which was normalised to *Renilla* luciferase activity.

## Chromatin immunoprecipitation assay

The ChIP assay was performed using a ChIP-IT Express Chromatin Immunoprecipitation Kit (Active Motif, Carlsbad, CA, USA) according to the manufacturer's protocol. Upon reaching 90% confluence, mIMCD3 cells were incubated with an isosmotic (300 mOsm) or hyperosmotic solution (600 mOsm) for 12 h, followed by immunoprecipitation at 4°C overnight with anti-NF-κB antibody (R&D Systems) or normal rabbit IgG as a control. The precipitated DNA was analysed by RT-PCR using the following primers: 5′-GCA CCA GGT CAA ACT CGT T-3′ (forward primer) and 5′-CCT GGA GCT TTC TCT ACA G-3′ (reverse primer). The amplified product (201 bp) was examined by electrophoresis and then sequenced. Quantitative statistics of the purified DNA precipitated by NF-κB were used for enrichment fold statistics using real-time PCR.

## Transfection and adenovirus infection

mIMCD3 cells were transfected with *Slc38a2* siRNA or mouse *Slc38a2*-C-EGFP plasmid using Lipofectamine 3000 reagent (Invitrogen) according to the manufacturer's instructions. *Slc38a2* siRNA was purchased from GenePharma (Shanghai, China). Specific siRNA sequences are listed below. The primers used for *Slc38a2*-mus-1704 were 5′-CUG CCA UGC UGA UCU UUA UTT-3′ (sense) and 5′-AUA AAG AUC AGC AUG GCA GTT-3′ (antisense). A full-length cDNA sequence (NM_175121) of mouse *Slc38a2* was cloned into the vector pex1-C-EGFP to express an EGFP-SLC38A2 fusion protein, hereafter referred to as the *Slc38a2*-C-EGFP plasmid. To overexpress SLC38A2, the cells were infected with an adenovirus carrying a cDNA encoding mouse full-length *Slc38a2* (Ad-SLC38A2) or an adenovirus carrying a cDNA encoding *Gfp* (Ad-GFP) for 24–36 h at a multiplicity of infection sufficient to infect >95% of the cells.

## Cell viability assay

Cell viability was determined using the MTT assay as previously described (*Zheng et al., 2014*). mIMCD3 cells were grown to 70–80% confluence in 24-well plates and subjected to hyperosmotic stress for an indicated period of time in the presence or absence of Ad-SLC38A2 ferroptosis agonists and inhibitors. MTT solution (5 mg/mL) was then added to the medium to a final concentration of 0.5 mg/mL. The cells were then cultured for another 2–4 h. At the end of the incubation period, the medium was carefully removed, and the formed MTT formazan crystals were dissolved by incubation in 500 μL of DMSO at 25°C in the dark for 15 min. Finally, the absorbance was measured at 490 nm using a microplate reader (TECAN, Mannedorf, Switzerland).

## Flow cytometry analysis of lipid ROS production

mIMCD3 cells were seeded in six-well plates, treated with the indicated compounds, and then subjected to hyperosmotic treatment. The cells were harvested after trypsinisation, resuspended in 500 μL of PBS containing 10 μM C11-BODIPY (581/591) (Cat #D3861; Invitrogen), and incubated for 30 min at 37°C in a tissue culture incubator. The cells were then resuspended in 500 μL of fresh PBS containing 2.5% BSA, strained through a 40 μm cell strainer, and analysed using a flow cytometer (FACSuite; BD Biosciences) equipped with a 488 nm laser for excitation.

## Total glutathione peroxidase detection

The cells were lysed on ice and then centrifuged at 12,000 ×*g* for 10 min at 4°C to collect the supernatant, which was then used to determine the enzymatic activity. The GPx activity of the cells was determined using a Total Glutathione Peroxidase Assay Kit (S0059S; Beyotime), according to the manufacturer's protocol. The protein concentration of the cells was determined using a Bradford dye-binding assay kit (P0006; Beyotime) (*Bradford, 1976*).

## Glutathione measurement in tissues

Renal medullary tissue was frozen in liquid nitrogen and ground into a powder. For each 10 mg of ground tissue powder, 30 μL of the protein removal reagent M solution was added, and the mixture was vortexed. Next, an additional 70 μL of protein removal reagent M solution was added, and the

mixture was fully homogenised using a glass homogeniser. The samples were then placed at 4°C for 10 min, followed by centrifugation at 10,000 ×g at 4°C for 10 min. The supernatant was collected, and the levels of glutathione reductive (GSH) and oxidised (GSSG) were determined using a commercial assay kit (S0053; Beyotime) according to the manufacturer's protocol.

### Iron assay

Tissue samples were obtained after the animals were perfused with normal saline (0.9% NaCl, containing 0.16 mg/mL heparin sodium) to clear the blood. The renal medulla was dissected, frozen in liquid nitrogen, and ground into powder. For each 100 mg of ground tissue powder, 900 μL of iron assay buffer was added for homogenisation, and then the mixture was centrifuged at 10,000 ×g for 10 min to remove insoluble cellular debris. Protein concentration in the tissue was determined using a Pierce BCA kit (Thermo Fisher Scientific). The supernatant was used to measure $Fe^{2+}$ levels using an assay kit (K773; Elabscience) according to the manufacturer's protocol.

### Measurement of lipid peroxidation

Renal medullary tissues were lysed in PBS, and the lysates were centrifuged at 12,000 ×g for 10 min at 4°C to collect the supernatant. The protein concentration of the supernatant was determined using a Pierce BCA kit (Thermo Fisher Scientific). MDA levels in the supernatant were measured using a kit (S0131S; Beyotime) according to the manufacturer's protocol.

### Measurement of total SOD activity

Renal medullary tissue samples were homogenised at 4°C in an SOD sample preparation solution and then centrifuged at 12,000 ×g at 4°C for 3–5 min. The supernatant was collected and used to measure the SOD activity with a kit (S0101S; Beyotime) according to the manufacturer's instructions.

### Measurement of caspase-3 activity

The activity of caspase-3 in renal medulla tissues was determined using the Caspase-3 Activity Assay Kit (C1116; Beyotime) according to the manufacturer's instructions. Briefly, after homogenisation of renal medulla tissue in cell lysis buffer, homogenates were centrifuged for 10 min at 16,000 ×g, and the supernatant was incubated with Ac-DEVD-pNA and reaction buffer for 90 min at 37°C. Caspase-3 activity was quantified in the samples at an absorbance of 405 nm according to the manufacturer's protocol. Protein concentrations in the tissue samples were determined using the Bradford method (P0006; Beyotime).

### Statistical analysis

Data are presented as the mean ± SEM. Statistical analyses were performed using the GraphPad Prism 8.0 software. All experiments were performed with a minimum of three biological replicates; the number of replicates is indicated in the figure legends. The details of the statistical tests are indicated in the figure legends. $p<0.05$ was regarded as the standard of significant difference.

## Acknowledgements

This work was supported by the National Natural Science Foundation of China Grants 82270703 (to XY), 81970606 (to XY), and 81970595 (to YF); the National Key R&D Program of China (2020YFC2005000); and the East China Normal University Medicine and Health Joint Fund (2022JKXYD03001).

## Additional information

### Funding

| Funder | Grant reference number | Author |
| --- | --- | --- |
| National Natural Science Foundation of China | 82270703 | Xiaoyan Zhang |

| Funder | Grant reference number | Author |
|---|---|---|
| National Natural Science Foundation of China | 81970606 | Xiaoyan Zhang |
| National Natural Science Foundation of China | 81970595 | Youfei Guan |
| National Key Research and Development Program of China | 2020YFC2005000 | Youfei Guan |
| East China Normal University | 2022JKXYD03001 | Xiaoyan Zhang |

The funders had no role in study design, data collection and interpretation, or the decision to submit the work for publication.

## Author contributions

Chunxiu Du, Data curation, Software, Formal analysis, Validation, Investigation, Methodology, Writing – original draft; Hu Xu, Supervision, Validation, Investigation, Methodology; Cong Cao, Jiahui Cao, Yufei Zhang, Cong Zhang, Rongfang Qiao, Wenhua Ming, Yaqing Li, Investigation; Huiwen Ren, Software; Xiaohui Cui, Data curation; Zhilin Luan, Formal analysis; Youfei Guan, Conceptualization, Supervision, Funding acquisition, Writing – review and editing; Xiaoyan Zhang, Supervision, Funding acquisition, Writing – review and editing

## Author ORCIDs

Chunxiu Du http://orcid.org/0000-0003-4152-4663
Hu Xu http://orcid.org/0000-0003-1198-0932
Yufei Zhang http://orcid.org/0000-0001-6289-8758
Huiwen Ren http://orcid.org/0000-0002-6037-8561
Youfei Guan http://orcid.org/0000-0002-5231-0209
Xiaoyan Zhang http://orcid.org/0000-0002-4060-2423

## Ethics

The use of animals and the study protocols were reviewed and approved by the Animal Care and Use Review Committee of Dalian Medical University and the study conformed to the Guide for the Care and Use of Laboratory Animals published by the US National Institutes of Health (in Guide for the Care and Use of Laboratory Animals, th, Editor. 2011: Washington).

## Decision letter and Author response

Decision letter https://doi.org/10.7554/eLife.80647.sa1
Author response https://doi.org/10.7554/eLife.80647.sa2

# Additional files

## Supplementary files

• MDAR checklist

• Supplementary file 1. Primer pairs used for real-time PCR to determine the mRNA levels of *Slc38a2*.

## Data availability

Sequencing data have been deposited in GEO under accession codes GSE206476. All data analysed during this study are included in the manuscript and supporting file; Source Data files have been provided for Figures and Figure supplements.

The following dataset was generated:

| Author(s) | Year | Dataset title | Dataset URL | Database and Identifier |
|---|---|---|---|---|
| Du C, Guan Y, Zhang X | 2022 | Neutral amino acid transporter SNAT2 protects renal medulla from hypertonicity-induced ferroptosis | https://www.ncbi.nlm.nih.gov/geo/query/acc.cgi?acc=GSE206476 | NCBI Gene Expression Omnibus, GSE206476 |

The following previously published dataset was used:

| Author(s) | Year | Dataset title | Dataset URL | Database and Identifier |
|---|---|---|---|---|
| Ransick A | 2019 | Single-Cell Profiling Reveals Sex, Lineage, and Regional Diversity in the Mouse Kidney | https://cello.shinyapps.io/kidneycellexplorer/ | NCBI Gene Expression Omnibus, GSE129798 |

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
