## [Editor Report]

These are nicely done and very complete studies which provide strong evidence that medullary collecting duct cells survive periodic medullary toxicity through upregulation of SLC38A2 , an amino acid transporter which provides substrate for glutathione production in the cells, which is in turn, protective. Cell lines, primary cultures and whole animals are used effectively to strengthen the mechanistic narrative.

---

## [Decision Letter]

**Decision letter after peer review:**

Thank you for submitting your article "Neutral amino acid transporter SLC38A2 protects renal medulla from hypertonicity-induced ferroptosis" for consideration by *eLife*. Your article has been reviewed by 3 peer reviewers, one of whom is a member of our Board of Reviewing Editors, and the evaluation has been overseen by Martin Pollak as the Senior Editor. The reviewers have opted to remain anonymous.

Essential revisions:

Reviewer #1. These are very well done studies which make the point convincingly that SLC38A2 plays an important role in protecting IMCD cells from hypertonic stress. The studies are complete and include cultured cell lines, primary cultures and intact mice. I have made a few suggestions about presentation in the public comments. The only consequential question is whether, in the intact mouse, water deprivation alters the upstream regulator (nF-kappaB) and the downstream TORC1 pathway as would be suggested by the studies in the cultured cells.

(1) For clarity, please consider moving the in vivo studies up in the results. This will motivate the detailed mechanistic studies in the cultured cells. The way the text is now, the reader wonders whether all of the work in cultured cells relates at all to the intact animal. You have such strong data in the intact mouse, that it likely should be moved up earlier in the narrative.

(2) Since the in vivo results correlate so well with the studies in culture, does water deprivation increase nF-kappaB or TORC1 expression in the intact mouse kidney?

Reviewer #2. I recommend another thorough proof-read by a native English speaker.

The terms hyperosmolarity and hypertonicity are used to describe the same experimental condition (hyperosmolarity). I suggest to consistently stick with the term "hyperosmolarity" to avoid confusion.

Slc38a2 and SLC38A2 are used to describe the same gene/protein. I recommend to use consistent nomenclature (Slc38a2) to avoid confusion.

Figure 7I does not contribute to the key message of this figure. It would be better to include these data into Figure 6.

Figure 8A, B were copied and pasted from https://cello.shinyapps.io/kidneycellexplorer/. Although these data support the expression of Slc38a2 mRNA in medullary cells, I suggest to move them into Supplementary Data.

Figure 8C. Detection of Slc38a2 protein in proximal tubules is surprising and non-consistent with the single cell sequencing data presented in Figure 8A and B. Is this staining specific or background? Control staining with secondary AB only would address this question.

Figure 9M: Increased TUNEL staining is suggested to be present in renal medullas of Slc38a2 knockout mice compared to control mice. This is an important finding. However, only two images of fluorescent stainings are presented. TUNEL positive cells should be quantitated and evaluated statistically.

Figure S9: Knockout mice excrete slightly increased urinary volume, but at the same time they display increased urinary osmolarity. This combination of findings is surprising and unexpected. Does this mean that knockout mice excrete more osmolytes than control mice? I would have expected equal osmolyte excretion rates. The authors should discuss this finding.

Reviewer #3. Show the functional significance of SLC38A2 in vivo by comparing the long-term effects of water deprivation (with and without mTor inhibitor) of WT and ko-mice.

Explain Tubular gain of function (i.e. increased urine osmolality) with SLC38A2 ko.

Electron microscopic evidence for ferroptosis in vivo.

Explain: mOsmo -SLC38A2/NFkB dose response in IMCD3 cells in vivo.

*Reviewer #1 (Recommendations for the authors):*

These studies address the mechanisms by which medullary collecting duct cells survive the rigors of periodic medullary hypertonicity which is needed for urinary concentration. The authors make a convincing case that hypertonicity leads to upregulation of the nF-kappaB pathway, which drives synthesis of the amino acid transporter, SLC38A2, which in turn provides critical amino acid substrates for the synthesis of the oxygen scavenger, glutathione. The studies combine work in cell lines, primary cultures and in the intact mouse. The work starts with an unbiased RNASeq approach which identified the upregulation of the SLC38A2 gene, and the authors then examine, systematically the role of upregulation of SLC38A2 in cell survival, and the pathways involved in SLC38A2 regulation. The studies are convincing and well performed.

I have only a few comments, which might improve the presentation.

1. Towards the end of the Results the authors turn to the whole animal, showing that water deprivation upregulates SLC38A2 expression, and showing the localization of SLC38A2 on cell membranes. I might move from the RNASeq to a description of the in vivo results in mice, including the knockout studies. These establish the importance of SLC38A2 in the intact mouse. I would then go on to use the cell culture studies to define the mechanisms involved.

2. In the intact mouse, does hypertonicity activate nF-KappaB (upstream stimulator of SLC38A2 synthesis) or TORC1 (downstream pathway putatively upregulated by SLC38A2)?

3. The SLC38A2 -/- mouse has (p 22) polyuria and "increased urine osmotic pressure." This term is unclear. Do the authors mean, hypotonic urine?

*Reviewer #2 (Recommendations for the authors):*

The authors use a mouse inner medullary collecting duct cell line (mIMCD-3), primary mouse medullary collecting duct (MCD) cells and a genetic mouse model to provide evidence that

(1) Hyperosmolarity induces cell death in mIMCD-3 cells that displays features of ferroptosis (lipid peroxidation, inhibition by ferroptosis inhibitors, potentiation by ferroptosis inducers)

(2) Slc38a2 (=SLC38A2), a previously identified neutral amino acid transporter providing substrates for intracellular glutathion (GSH) production, is strongly induced by high osmolarity in mIMCD-3 and primary MCD cells.

(3) Activation of nuclear factor kappa B (NF-kappaB) in response to hyperosmolarity is an important upstream regulator of Slc38a2 in mIMCD-3 cells.

(4) Overexpression of Slc38a2 by adenovirus inhibits ferroptosis and that inhibition or downregulation of Slc38a2 aggravates hypertonicity-induced ferroptosis in mIMCD3 and primary IMCD cells.

(5) Slc38a2 inhibits ferroptosis under hypertonic stress in mIMCD-3 cells by activating mTOR signaling.

(6) Slc38a2 is expressed in medullary epithelial cells of the mouse nephron and its mRNA and protein is induced in response to water deprivation.

(7) Water deprived Slc38a2 knockout mice display evidence of ferroptotic cell death in the renal medulla.

Strengths:

The study is thoroughly conducted and presented well.

Extensive in vitro studies are provided using a mouse inner medullary collecting duct cell line and primary mouse medullary collecting duct cells.

Novel molecular insights into hyperosmolarity-induced regulation of Slc38a2 in collecting duct cells are provided.

Novel molecular insights into the relationship of Slc38a2 and the prevention of ferroptosis in collecting duct cells are provided.

in vivo evidence from Slc38a2 knockout mice largely supports the in vitro findings.

Weaknesses:

No major weaknesses are identified.

Conclusion:

The authors provide important and novel information on the molecular basis of osmotic resistance in the renal medulla. The results are carefully presented and interpreted. The data will be important to the field of renal physiology and might be of relevance to the molecular basis of human disease.

*Reviewer #3 (Recommendations for the authors):*

In their Du et al. describe that hypertonicity induces regulated cell death of IMCD3 and primary renal medullary collecting cells via ferroptosis. They show that hypertonicity induces the neutral amino acid transporter SLC38A2 in these cells and conversely that SLC38A2 depletion or inhibition further increase ferroptosis. These findings are paralleled by in vivo studies showing induction of SLC38A2 in medullary collecting cells and other tubule segments and increased TUNEL signal following water dehydration.

Strengths:

The authors put up a model explaining the role of SLC38A2 in the protection of IMCD3 and primary renal medullary collecting cells against hypertonicity, which is supported by the effects of SLC38A2 induction by hypertonicity (mRNA and protein level) in the above cells in vitro. It is further supported by the use of various inhibitors and enhancers of ferroptosis. The authors also show induction of SLC38A2 by NFkB. in vivo, tubular (including IMC ducts) SLC38A2 expression is induced by water deprivation, which is in line with their model. Increased numbers of TUNEL positive cells are observed in SLC38A2 ko mice, consistent with their in vitro data and their model.

Weakness:

The authors demonstrate maximal induction of SLC38A2 mRNA and protein and 600 mOsm. However, they also show declining levels SLC38A2 mRNA and protein at 700 and 800 mOsm (Figure 3). The authors only describe the time course of induction but not describe the dose response curve. As urine osmolality can go much higher the question arises why SLC38A2 should be considered protective when its is not expressed in IMCD3 cells with 700 mOsm and higher. So what is the role of SLC38A2 above 600 mOsm?

Figure S3: Induction of Slco4a1 and Slc5a3 mRNA was much stronger with hyperosmolality compared to the effect on Slc38a2 mRNA. Can the authors exclude a role of the former in protection against hyperosmolality?

Figure 4: The authors do not explain why they study NFkB. They show that hypertonicity dose-dependently increased NF-κB activity in mIMCD3 cells from 300-800 mOsm. How do the authors reconcile this notion with repression of SLC38A2 expression with 700 and 800 mOsm (Figure 3.)?

Figure 9: Can the authors provide independent evidence for ferroptosis by performing electron microscopic studies of renal medullae?

Figure S9: The authors demonstrate increased urine osmolality in SLC38A2 ko mice (despite higher urine output). How do the authors explain the obviously increased medullary osmotic gradient?

How do the authors explain the "polyuria" under basal (S9b) and challenged conditions (S9e).

Significantly increased cell death should rather decrease but not increase the medullary osmotic gradient at least during longer term experiments. Such experiments (maybe combined with Tor-inhibitors) may yield insight into the functional role of SLC38A2 in water deprived animals.

In the discussion, the authors state: "Finally, by using SLC38A2 gene knockout mice, we confirm the critical role of SLC38A2 in maintaining renal medullary homeostasis and renal urine concentration by attenuating medullary cell ferroptosis." The increased urine osmolality in SLC38A2 ko mice indicates the opposite.

The authors use different osmolarities ranging between 300-1200 mOsm without explaining why.

(i.e. Figure 1a-c with 300 and 600 mOSM vs. With 300 and 500 mOSM in Figure 1d-f).

Figure 9 M: Can the authors localize TUNEL-positivity to either interstitial and/or tubular cells?

---

## [Author Response]

Essential revisions:Reviewer #1. These are very well done studies which make the point convincingly that SLC38A2 plays an important role in protecting IMCD cells from hypertonic stress. The studies are complete and include cultured cell lines, primary cultures and intact mice. I have made a few suggestions about presentation in the public comments. The only consequential question is whether, in the intact mouse, water deprivation alters the upstream regulator (nF-kappaB) and the downstream TORC1 pathway as would be suggested by the studies in the cultured cells.

We appreciate the reviewer for the positive comments on our present work.

(1) For clarity, please consider moving the in vivo studies up in the results. This will motivate the detailed mechanistic studies in the cultured cells. The way the text is now, the reader wonders whether all of the work in cultured cells relates at all to the intact animal. You have such strong data in the intact mouse, that it likely should be moved up earlier in the narrative.

Thank you for the meaningful comment. According to your suggestion, we have moved the in vivo studies up in the result session in the revised manuscript.

(2) Since the in vivo results correlate so well with the studies in culture, does water deprivation increase nF-kappaB or TORC1 expression in the intact mouse kidney?

Thank you for the valuable question. As suggested by the reviewer, we have measured the levels of NF-κB and mTORC1. As shown in Figure 6C in the revised manuscript, water deprivation upregulated the p-NF-κB expression in the renal medullas of wild-type mice, suggesting that the NF-κB pathway is activated in dehydrated mouse kidneys. In addition, water deprivation markedly downregulated the p-S6 and p-4E-BP1 expression in renal medullas, indicating that the mTORC1 pathway is suppressed in the kidneys of dehydrated mice (please see Author response image 1).

**Author response image 1. sa2fig1:** Effect of water restriction on renal medullary mTOC1 activity. Water deprivation significantly decreased p-S6 and p-4E-BP1 protein levels in renal medullas of wild-type C57BL/6 mice. The mice were subjected to water deprivation (WD) for 24 h or free access to water (Control).

Reviewer #2.I recommend another thorough proof-read by a native English speaker.

Thank you very much for your suggestion. We have invited a professional native English speaker to edit our manuscript.

The terms hyperosmolarity and hypertonicity are used to describe the same experimental condition (hyperosmolarity). I suggest to consistently stick with the term "hyperosmolarity" to avoid confusion.

We sincerely thank the reviewer for your careful reading. As suggested, we have replaced the term “hypertonicity” by “hyperosmolarity” in the revised manuscript to avoid confusion.

Slc38a2 and SLC38A2 are used to describe the same gene/protein. I recommend to use consistent nomenclature (Slc38a2) to avoid confusion.

Thank you very much for your suggestion. As the reviewer suggested, we have changed the word “SLC38A2” to “Slc38a2” in the entire manuscript.

Figure 7I does not contribute to the key message of this figure. It would be better to include these data into Figure 6.

Thank the reviewer for the valuable suggestion. As suggested, we have moved the Figure 7I to Figure 6J in the original paper, which is now shown in new Figure 8J in the revised manuscript.

Figure 8A, B were copied and pasted from https://cello.shinyapps.io/kidneycellexplorer/. Although these data support the expression of Slc38a2 mRNA in medullary cells, I suggest to move them into Supplementary Data.

Thank you for the constructive suggestion. In the revised manuscript, we have moved the Figure 8A-B into Supplementary Data as Figure 3—figure supplement 1.

Figure 8C. Detection of Slc38a2 protein in proximal tubules is surprising and non-consistent with the single cell sequencing data presented in Figure 8A and B. Is this staining specific or background? Control staining with secondary AB only would address this question.

Thank you for the valuable comments and suggestions. We routinely used non-specific IgG as a negative control. The results showed that staining for SLC38A2 protein was specific (Author response image 2). The detection of SLC38A2 protein in the proximal tubules appears not to be consistent with the finding presented in Figure 8A and B (Figure 3—figure supplement 1 in the revised manuscript) in which the single cell sequencing data showed a very low mRNA level of *Slc38a2* in this renal tubular segment. The reason for this inconsistence is not clear. We speculate that some post-transcriptional and post-translational mechanisms may be involved in the regulation of the stability of SLC38A2 mRNA and protein. This important issue warrants further investigation.

**Author response image 2. sa2fig2:** Intrarenal localisation of SLC38A2 protein in the kidney. Immunohistochemical study indicating intrarenal localisation of SLC38A2 protein in mouse. The results showed that SLC38A2 was expressed at relatively high level in the proximal tubule, thick ascending limb, and collecting duct, with low expression in the distal tubule. AQP1, THP, NCC, AQP2 are markers for the proximal tubule, distal tubule, thick ascending limb and collecting duct, respectively. IgG was used as a negative control.

Figure 9M: Increased TUNEL staining is suggested to be present in renal medullas of Slc38a2 knockout mice compared to control mice. This is an important finding. However, only two images of fluorescent stainings are presented. TUNEL positive cells should be quantitated and evaluated statistically.

Thank you for the valuable suggestion. As the reviewer suggested, the TUNEL positive cells have been quantitated and evaluated statistically. The results are now included in Figure 4N in the revised manuscript (Figure 9M in the original paper).

Figure S9: Knockout mice excrete slightly increased urinary volume, but at the same time they display increased urinary osmolarity. This combination of findings is surprising and unexpected. Does this mean that knockout mice excrete more osmolytes than control mice? I would have expected equal osmolyte excretion rates. The authors should discuss this finding.

Thank you for the constructive comment and suggestion. As shown in Figure S9 (Figure 4—figure supplement 2 in the revised manuscript), the *Slc38a2*-knockout mice indeed excreted slightly increased urinary volume. As described in the manuscript, SLC38A2 is an important neutral amino acid transporter in vivo and amino acids can serve as important osmolytes. We measured urine osmolyte and amino acid excretion in both wild-type (WT) and the *Slc38a2*-knockout (KO) mice. The results showed that daily osmolyte and amino acid excretion in the *Slc38a2*-KO mice was significantly higher than that in WT mice (Author response image 3 and B), suggesting a slight increase in urinary osmolality. It is speculated that as an amino acid transporter, *Slc38a2*-gene deficiency results in blunted cellular uptake of certain amino acids by peripheral tissues and elevated level of amino acids in the plasma, which causes a mild osmotic diuresis. In addition, under dehydrate condition, increased collecting duct ferroptosis as reported in this paper may also contribute to increased urine output in the *Slc38a2*-knockout mice. In the revised manuscript, we briefly discussed this important issue.

**Author response image 3. sa2fig3:** Daily urinary excretion of osmotic solutes and amino acids was increased in the Slc38a2- KO mice. (A) Urine was collected from wild type (WT) and the *Slc38a2*-knockout (KO) mice (8-10 weeks old, male) for 24 h in the urine metabolic cage. The *Slc38a2*-KO mice exhibited more urine osmolyte excretion than WT mice. Urine osmotic pressure and urine volume were used to detect urine osmotic solutes (urine volume multiplied by urine osmotic pressure). (B) The *Slc38a2*-KO mice excreted more total amino acids in the urine than WT mice. Total Amino Acids (T-AA) Colorimetric Assay Kit was used to measure total amino acids in mouse urine (Elabscience, K055-M). Data are means ± SEM; two-tailed Student’s t test was used. P<0.01, n=6-9.

Reviewer #3. Show the functional significance of SLC38A2 in vivo by comparing the long-term effects of water deprivation (with and without mTor inhibitor) of WT and ko-mice.

Thank the reviewer for the meaningful suggestion. Since long-term water deprivation is very harmful for experimental animals (PMID: 31818909; PMID: 17536615), the Animal Ethics Committee only allowed us to extend the period of water deprivation (WD) to 48 hours. Wild-type (WT) and the *Slc38a2*-knockout (KO) mice were divided into following groups: WT-WD-vehicle (n=9), KO-WD-vehicle (n=7), WT-WD-rapamycin (n=11), and KO-WD-rapamycin (n=8). The mice were intraperitoneally injected with rapamycin (2mg/kg/day) or vehicle (saline) for 7 consecutive days and were subjected to 2-day water deprivation staring from the fifth day. The results showed that both WT and KO mice receiving saline treatment can tolerate 2-day water restriction. However, in the presence of rapamycin, almost all KO mice were dead after 2-day water deprivation. Although WT mice survived 7-day rapamycin treatment and 2-day water restriction, all of them were in serious health condition. Therefore, we decided to terminate the experiment and could not complete the study. However, the findings further provided evidence supporting the importance of SLC38A2 (SLC38A2) in maintaining water homeostasis.

Explain Tubular gain of function (i.e. increased urine osmolality) with SLC38A2 ko.

Thank you for the meaningful comment. It is well known that SLC38A2 is an important neutral amino acid transporter in vivo and amino acids can serve as important osmolytes. We measured urine osmolyte and amino acid excretion in both wild-type (WT) and the *Slc38a2*-knockout (KO) mice. The results showed that daily osmolyte and amino acid excretion in the *Slc38a2*-KO mice was significantly higher than that in WT mice (Author response image 3 and B), suggesting a slight increase in urinary osmolality. As an amino acid transporter, *Slc38a2*-gene deficiency results in blunted cellular uptake of certain amino acids by peripheral tissues and elevated levels of amino acids in the plasma (data not shown), which causes a mild osmolytic diuresis. Therefore, we speculate that increased urine osmolyte excretion may be due to increased glomerular filtration of amino acids into renal tubules, rather than a result of tubular gain of function. In the present study, we provide clear evidence that under dehydrate condition, collecting duct cell ferroptosis was significantly higher in the *Slc38a2*-knockout mice than that in wild-type cells. In the revised manuscript, we briefly discussed this important issue.

Electron microscopic evidence for ferroptosis in vivo.

Thank you for this important comment. As the reviewer suggested, we conducted an electron microscopic study on renal medullas of wild-type (WT) and the *Slc38a2*-knockout (KO) mice with 24-hour water deprivation. The results showed although both genotypes exhibited robust changes in mitochondrial morphology, the reduction of mitochondria volume and loss and rupture of mitochondrial crest were more severe in medullary cells of the KO mice than that in WT mice under water deprivation condition, which was in line with the findings in cultured cells under hyperosmolarity. Therefore, both in vitro and in vivo studies demonstrate that hyperosmolarity can aggravate medullary cell ferroptosis (Author response image 4).

**Author response image 4. sa2fig4:** Electron microscope examination showed that ferroptosis of renal medullary cells was aggravated in the *Slc38a2*-KO mice under water deprivation condition. The renal medullas of mice were collected and analyzed by electron microscope after 24-h water deprivation. Representative images showed that more severe medullary cell ferroptosis as reflected by the reduction of mitochondria volume and loss and rupture of mitochondrial crest (arrows) was found in the KO mice than that in WT mice under water deprivation. Scale bar = 0.5μm. WD: water deprivation.

Explain: mOsmo -SLC38A2/NFkB dose response in IMCD3 cells in vivo.

Thanks for the valuable comment. We guess the reviewer means dose response in mIMCD3 in vitro, since mIMCD3 is a mouse collecting duct cell line. It has been well known that NF-κB as a transcription factor plays an important role in promoting the survival of the collecting duct cells. A recent study (PMID: 32175843) showed that the induction of SLC38A2 (SLC38A2) is mediated by the nuclear factor κB (NF-κB) family protein, which is consistent with our findings. In the mIMCD3 cells, we found a dose-dependent response of active NF-κB (p-NF-κB) to osmotic pressure between 300 mOsm and 800 mOsm (Figure 6A). However, in terms of SLC38A2, we only observed a dose-dependent response of SLC38A2 protein between 300 mOsm and 600 mOsm (Figure 5B). The reason for the inconsistence remains unclear. One possibility is that the half-life of SLC38A2 protein is shorter than that of NF-κB protein. The other possibility is that the effect of NF-κB on SLC38A2 expression is dependent on the NF-κB activity. At low level NF-κB induces SLC38A2 expression, while at high level it suppresses SLC38A2 expression. However, the precise mechanism by which NF-κB affects SLC38A2 expression requires further investigation.

Reviewer #1 (Recommendations for the authors):These studies address the mechanisms by which medullary collecting duct cells survive the rigors of periodic medullary hypertonicity which is needed for urinary concentration. The authors make a convincing case that hypertonicity leads to upregulation of the nF-kappaB pathway, which drives synthesis of the amino acid transporter, SLC38A2, which in turn provides critical amino acid substrates for the synthesis of the oxygen scavenger, glutathione. The studies combine work in cell lines, primary cultures and in the intact mouse. The work starts with an unbiased RNASeq approach which identified the upregulation of the SLC38A2 gene, and the authors then examine, systematically the role of upregulation of SLC38A2 in cell survival, and the pathways involved in SLC38A2 regulation. The studies are convincing and well performed.I have only a few comments, which might improve the presentation.1. Towards the end of the Results the authors turn to the whole animal, showing that water deprivation upregulates SLC38A2 expression, and showing the localization of SLC38A2 on cell membranes. I might move from the RNASeq to a description of the in vivo results in mice, including the knockout studies. These establish the importance of SLC38A2 in the intact mouse. I would then go on to use the cell culture studies to define the mechanisms involved.

Thank you for this important comment. According to your suggestion, we reorganized the paper by showing in vivo results first and then in vitro findings in the revised manuscript.

2. In the intact mouse, does hypertonicity activate nF-KappaB (upstream stimulator of SLC38A2 synthesis) or TORC1 (downstream pathway putatively upregulated by SLC38A2)?

Thanks for the meaningful question. In the present study, we found that in wild-type mice water deprivation upregulated the NF-κB pathway in renal medullas (please see Figure 6C in the revised manuscript), but downregulated the mTORC1 pathway (Author response image 1) which is consistent with our in vitro finding.

3. The SLC38A2 -/- mouse has (p 22) polyuria and "increased urine osmotic pressure." This term is unclear. Do the authors mean, hypotonic urine?

Thanks a lot for your careful reading and comment. Indeed, the term “increased urine osmotic pressure” is not appropriate. What we meant is that the SLC38A2-/- (*Slc38a2^-/-^*) mice exhibited increased urine osmolyte excretion. As described in the manuscript, SLC38A2 is an important neutral amino acid transporter in vivo and amino acids can serve as important osmolytes. We measured urine osmolyte and amino acid excretion in both wild-type (WT) and *Slc38a2*-knockout (KO) mice. The results showed that daily osmolyte and amino acid excretion in the *Slc38a2*-KO mice was significantly higher than that in WT mice, suggesting a slight increase in urinary osmolality. We have corrected the term in the revised manuscript.

Reviewer #2 (Recommendations for the authors):The authors use a mouse inner medullary collecting duct cell line (mIMCD-3), primary mouse medullary collecting duct (MCD) cells and a genetic mouse model to provide evidence that(1) Hyperosmolarity induces cell death in mIMCD-3 cells that displays features of ferroptosis (lipid peroxidation, inhibition by ferroptosis inhibitors, potentiation by ferroptosis inducers)(2) Slc38a2 (=SLC38A2), a previously identified neutral amino acid transporter providing substrates for intracellular glutathion (GSH) production, is strongly induced by high osmolarity in mIMCD-3 and primary MCD cells.(3) Activation of nuclear factor kappa B (NF-kappaB) in response to hyperosmolarity is an important upstream regulator of Slc38a2 in mIMCD-3 cells.(4) Overexpression of Slc38a2 by adenovirus inhibits ferroptosis and that inhibition or downregulation of Slc38a2 aggravates hypertonicity-induced ferroptosis in mIMCD3 and primary IMCD cells.(5) Slc38a2 inhibits ferroptosis under hypertonic stress in mIMCD-3 cells by activating mTOR signaling.(6) Slc38a2 is expressed in medullary epithelial cells of the mouse nephron and its mRNA and protein is induced in response to water deprivation.(7) Water deprived Slc38a2 knockout mice display evidence of ferroptotic cell death in the renal medulla.Strengths:The study is thoroughly conducted and presented well.Extensive in vitro studies are provided using a mouse inner medullary collecting duct cell line and primary mouse medullary collecting duct cells.Novel molecular insights into hyperosmolarity-induced regulation of Slc38a2 in collecting duct cells are provided.Novel molecular insights into the relationship of Slc38a2 and the prevention of ferroptosis in collecting duct cells are provided.

in vivo *evidence from Slc38a2 knockout mice largely supports the* in vitro *findings.*

Weaknesses:No major weaknesses are identified.Conclusion:The authors provide important and novel information on the molecular basis of osmotic resistance in the renal medulla. The results are carefully presented and interpreted. The data will be important to the field of renal physiology and might be of relevance to the molecular basis of human disease.

We appreciate the reviewer for the positive comments on our present work.

Reviewer #3 (Recommendations for the authors):In their Du et al. describe that hypertonicity induces regulated cell death of IMCD3 and primary renal medullary collecting cells via ferroptosis. They show that hypertonicity induces the neutral amino acid transporter SLC38A2 in these cells and conversely that SLC38A2 depletion or inhibition further increase ferroptosis. These findings are paralleled by in vivo studies showing induction of SLC38A2 in medullary collecting cells and other tubule segments and increased TUNEL signal following water dehydration.Strengths:The authors put up a model explaining the role of SLC38A2 in the protection of IMCD3 and primary renal medullary collecting cells against hypertonicity, which is supported by the effects of SLC38A2 induction by hypertonicity (mRNA and protein level) in the above cells in vitro. It is further supported by the use of various inhibitors and enhancers of ferroptosis. The authors also show induction of SLC38A2 by NFkB. in vivo, tubular (including IMC ducts) SLC38A2 expression is induced by water deprivation, which is in line with their model. Increased numbers of TUNEL positive cells are observed in SLC38A2 ko mice, consistent with their in vitro data and their model.Weakness:The authors demonstrate maximal induction of SLC38A2 mRNA and protein and 600 mOsm. However, they also show declining levels SLC38A2 mRNA and protein at 700 and 800 mOsm (Figure 3). The authors only describe the time course of induction but not describe the dose response curve. As urine osmolality can go much higher the question arises why SLC38A2 should be considered protective when its is not expressed in IMCD3 cells with 700 mOsm and higher. So what is the role of SLC38A2 above 600 mOsm?

Thank the reviewer for the meaningful comment. The mIMCD3 is a mouse inner medullary collecting duct cell line which is commonly used for studying the collecting duct function in vitro. The mIMCD3 cells were usually cultured in isotonic medium, which is in sharp contrast to medullary collecting duct cells in the kidney who live in a hypertonic and hypoxic environment. In addition, as a transformed cells, mIMCD3 cells loss many characteristics of renal medullary collecting duct cells. As a result, mIMCD3 cells can only tolerate hypertonicity up to 600 mOsm. In the hypertonic medium with 700 mOsm and higher, most of mIMCD3 cells are dead, leading to the degradation of most of proteins including SLC38A2 (SLC38A2). As the reviewer pointed out, urine and medullary osmolality can go much higher under dehydration condition. To better mimic the in vivo characteristics of medullary collecting duct cells, we also cultured primary renal collecting duct cells who can tolerate much higher osmolality. As shown in Figure 8E in the revised manuscript, SLC38A2 protein expression remains at high level under 1200 mOsm. Therefore, as described in the manuscript, SLC38A2 has an important protective effect on the survival of medullary cells under hypertonic conditions above 600 mOsm in vivo.

Figure S3: Induction of Slco4a1 and Slc5a3 mRNA was much stronger with hyperosmolality compared to the effect on Slc38a2 mRNA. Can the authors exclude a role of the former in protection against hyperosmolality?

The reviewer raised an important point. In the kidney, the protective role of *Slc5a3* against hypertonic stress has been repeatedly reported (PMID: 7948784; PMID: 8430828; PMID: 9685419). In terms of *Slco4a1*, although many reports demonstrate a role in tumorigenesis, its role in osmoprotection in the kidney remains largely unknown and is currently under investigation in our group.

Figure 4: The authors do not explain why they study NFkB. They show that hypertonicity dose-dependently increased NF-κB activity in mIMCD3 cells from 300-800 mOsm. How do the authors reconcile this notion with repression of SLC38A2 expression with 700 and 800 mOsm (Figure 3.)?

Thanks for the wonderful comment. It has been well known that NF-κB as a transcription factor plays an important role in promoting the survival of the collecting duct cells. A recent study (PMID: 32175843) showed that the induction of SLC38A2 (SLC38A2) is mediated by the nuclear factor κB (NF-κB) family protein, which is consistent with our findings. In the mIMCD3 cells, we found a dose-dependent response of active NF-κB (p-NF-κB) to osmotic pressure between 300 mOsm and 800 mOsm (Figure 6A). However, in terms of SLC38A2, we only observed a dose-dependent response of SLC38A2 protein between 300 mOsm and 600 mOsm (Figure 5B). The reason for the inconsistence remains unclear. One possibility is that the half-life of SLC38A2 protein is shorter than that of NF-κB protein. The other possibility is that the effect of NF-κB on SLC38A2 expression is dependent on the NF-κB activity. At low level NF-κB induces SLC38A2 expression, while at high level it suppresses SLC38A2 expression. However, the precise mechanism by which NF-κB affects SLC38A2 expression requires further investigation.

Figure 9: Can the authors provide independent evidence for ferroptosis by performing electron microscopic studies of renal medullae?

Thank you for this important comment. As the reviewer suggested, we conducted an electron microscopic study on renal medullas of wild-type (WT) and the *Slc38a2*-knockout (KO) mice with 24-hour water deprivation. The results showed that although both genotypes exhibited robust changes in mitochondrial morphology, the reduction of mitochondria volume and loss and rupture of mitochondrial crest were more severe in renal medullary cells of the KO mice than that in WT mice under water deprivation condition, which was in line with the findings in cultured cells under hyperosmolarity. Therefore, both in vitro and in vivo studies demonstrate that hyperosmolarity can aggravate renal medullary cell ferroptosis (Author response image 4).

Figure S9: The authors demonstrate increased urine osmolality in SLC38A2 ko mice (despite higher urine output). How do the authors explain the obviously increased medullary osmotic gradient?

Thank you for the constructive comment and suggestion. As shown in Figure S9 (Figure 4—figure supplement 2 in the revised manuscript), the *Slc38a2*-knockout mice indeed excreted slightly increased urinary volume. As described in the manuscript, SLC38A2 is an important neutral amino acid transporter in vivo and amino acids can serve as important osmolytes. We measured urine osmolyte and amino acid excretion in both wild-type (WT) and *Slc38a2*-knockout (KO) mice. The results showed that daily osmolyte and amino acid excretion in the *Slc38a2*-KO mice was significantly higher than that in WT mice (Author response image 3 and B), suggesting a slight increase in urinary osmolality. It is speculated that as an amino acid transporter, *Slc38a2*-gene deficiency results in blunted uptake of certain amino acids by peripheral tissues and elevated level of amino acids in the plasma, which causes a mild osmotic diuresis. In addition, under dehydrate condition, increased collecting duct ferroptosis as reported in this paper may also contribute to increased urine output in the *Slc38a2*-knockout mice. In terms of medullary osmotic gradient, we speculate that it is slightly reduced due to massive ferroptosis of medullary collecting duct cells in the *Slc38a2*-knockout mice after dehydration. In the revised manuscript, we briefly discussed this important issue.

How do the authors explain the "polyuria" under basal (S9b) and challenged conditions (S9e).

Thanks a lot for your careful reading and meaningful question. It is speculated that as an amino acid transporter, *Slc38a2*-gene deficiency results in blunted cellular uptake of certain amino acids by peripheral tissues and elevated level of amino acids in the plasma, which causes a mild osmotic diuresis. In addition, under dehydrate condition, increased collecting duct ferroptosis as reported in this paper may also contribute to increased urine output in the *Slc38a2*-knockout mice.

Significantly increased cell death should rather decrease but not increase the medullary osmotic gradient at least during longer term experiments. Such experiments (maybe combined with Tor-inhibitors) may yield insight into the functional role of SLC38A2 in water deprived animals.

Thank the reviewer for the meaningful suggestion. Since long-term water deprivation is very harmful for experimental animals (PMID: 31818909; PMID: 17536615), the Animal Ethics Committee only allowed us to extend the period of water deprivation (WD) to 48 hours. Wild-type (WT) and *Slc38a2*-knockout (KO) mice were divided into following groups: WT-WD-vehicle (n=9), KO-WD-vehicle (n=7), WT-WD-rapamycin (n=11), and KO-WD-rapamycin (n=8). The mice were intraperitoneally injected with rapamycin (2mg/kg/day) or vehicle (saline) for 7 consecutive days and were subjected to 2-day water deprivation staring from the fifth day. The results showed that both WT and KO mice receiving saline treatment can tolerate 2-day water restriction. However, in the presence of rapamycin, almost all KO mice were dead after 2-day water deprivation. Although WT mice survived 7-day rapamycin treatment and 2-day water restriction, all of them were in serious health condition. Therefore, we decided to terminate the experiment but could not complete the study. However, the findings further provided evidence supporting the importance of SLC38A2 (SLC38A2) in maintaining water homeostasis.

In the discussion, the authors state: "Finally, by using SLC38A2 gene knockout mice, we confirm the critical role of SLC38A2 in maintaining renal medullary homeostasis and renal urine concentration by attenuating medullary cell ferroptosis." The increased urine osmolality in SLC38A2 ko mice indicates the opposite.

Thank you for the valuable comment. In the present study, we showed that the *Slc38a2* (SLC38A2) gene knockout (KO) aggravated ferroptosis of medullary collecting duct cells, leading to an impaired urine concentration and increased urine output. The increased urine osmolyte excretion is possibly due to enhanced glomerular filtration of amino acids into renal tubules as a results of elevated plasma amino acid levels (data not shown). Another possibility is that *Slc38a2*-gene deficiency in renal tubules reduced the capacity of renal reabsorption of the filtrated amino acids, resulting in a slight increase in urine osmolality and mild osmotic diuresis.

The authors use different osmolarities ranging between 300-1200 mOsm without explaining why. (i.e. Figure 1a-c with 300 and 600 mOSM vs. With 300 and 500 mOSM in Figure 1d-f).

Thanks for your meaningful question. The mIMCD3 is a mouse inner medullary collecting duct cell line which is commonly used for studying the collecting duct function in vitro. The mIMCD3 cells were usually cultured in isotonic medium, which is in sharp contrast to medullary collecting duct cells in the kidney who live in a hypertonic and hypoxic environment. In addition, as a transformed cells, mIMCD3 cells loss many characteristics of renal medullary collecting duct cells after many passages. As a result, mIMCD3 cells can only tolerate hypertonicity up to 600 mOsm and in the hypertonic medium with 700 mOsm and higher, most of mIMCD3 cells are dead. As the reviewer mentioned, urine and medullary osmolality can go much higher in vivo under dehydration condition. To better mimic the in vivo characteristics of medullary collecting duct cells, we also cultured primary renal collecting duct cells who can tolerate much higher osmolality, which allowed us to determine the effect of higher osmotic pressure (up to 1200 mOsm) on these cells.

As mentioned above, the mIMCD3 cells are more suitable for testing the effect of osmolality between 300 mOsm and 800 mOsm. Since hypertonicity causes ferroptosis of mIMCD3 cells in a dose dependent manner, we chose 600 mOsm to determine the protective effect of the ferroptosis inhibitor and 500 mOsm to measure injurious effect of the ferroptosis agonist.

Figure 9 M: Can the authors localize TUNEL-positivity to either interstitial and/or tubular cells?

Thank you for the constructive comment. As the reviewer suggested, we co-stained TUNEL positive cells with AQP2, a marker for renal collecting duct cells. The results showed that the cells with positive tunnel staining (green) were co-localized with the collecting duct marker AQP2 (red) in renal medulla (Author response image 5). This finding demonstrates that the TUNEL positive cells were mainly the medullary collecting ducts rather than interstitial cells.

**Author response image 5. sa2fig5:** Increased TUNEL-positive cells in renal medullary collecting duct cells of the *Slc38a2*-KO mice after water deprivation. Immunofluorescence analysis showing that water restriction resulted in a marked increase in cell death (green) in the medullas of the *Slc38a2-/-* mice as assessed by the TUNEL assay. TUNEL staining (green) and AQP2 (red) staining showed that the TUNEL-positive cells were mainly renal medullary collecting ducts. AQP2: Aquaporin 2, a marker of renal collecting duct cells. WT: wild-type; KO: *Slc38a2-/-*; WD: water deprivation. Scale bar=25μm.